# Intrinsically stretchable organic photovoltaics by redistributing strain to PEDOT:PSS with enhanced stretchability and interfacial adhesion

Jiachen Wang [1,2,3], Yuto Ochiai [2], Niannian Wu[2,4], Kiyohiro Adachi [2], Daishi Inoue[2], Daisuke Hashizume [2], Desheng Kong [5], Naoji Matsuhisa [6,7], Tomoyuki Yokota [1,8], Qiang Wu[9], Wei Ma [9], Lulu Sun [10], Sixing Xiong [2], Baocai Du[1,2], Wenqing Wang[1,2], Chih-Jen Shih[3], Keisuke Tajima [2], Takuzo Aida [2,4], Kenjiro Fukuda [2,10] ✉ & Takao Someya [1,2,10] ✉

Intrinsically stretchable organic photovoltaics have emerged as a prominent candidate for the next-generation wearable power generators regarding their structural design flexibility, omnidirectional stretchability, and in-plane deformability. However, formulating strategies to fabricate intrinsically stretchable organic photovoltaics that exhibit mechanical robustness under both repetitive strain cycles and high tensile strains remains challenging. Herein, we demonstrate high-performance intrinsically stretchable organic photovoltaics with an initial power conversion efficiency of 14.2%, exceptional stretchability (80% of the initial power conversion efficiency maintained at 52% tensile strain), and cyclic mechanical durability (95% of the initial power conversion efficiency retained after 100 strain cycles at 10%). The stretchability is primarily realised by delocalising and redistributing the strain in the active layer to a highly stretchable PEDOT:PSS electrode developed with a straightforward incorporation of ION E, which simultaneously enhances the stretchability of PEDOT:PSS itself and meanwhile reinforces the interfacial adhesion with the polyurethane substrate. Both enhancements are pivotal factors ensuring the excellent mechanical durability of the PEDOT:PSS electrode, which further effectively delays the crack initiation and propagation in the top active layer, and enables the limited performance degradation under high tensile strains and repetitive strain cycles.

Organic photovoltaics (OPVs) have garnered attention as promising advanced wearable power generators for on-skin electronics, owing to their remarkably high power conversion efficiency (PCE) and flexibility[1–3]. To secure the structural and functional integrity of the epidermal power generators, sufficient mechanical robustness must be ensured for withstanding the repeated tensile strains imposed on the devices by continuous body movements, thereby permitting long-term reliable operation[4–7]. This is crucial because the cyclic strains generated under synchronous deformation with skin may induce structural vulnerabilities, thereby gradually deteriorating the long-

term performance[8,9]. Consequently, there is a pressing need to pursue the development of stretchable OPVs with both high PCE and good mechanical durability beyond the realm of flexibility.

Following the introduction of pre-stretching, a trailblazing structural engineering method realised by attaching an ultrathin device on a pre-stretched elastomer, that has enabled a transformative leap for OPVs from being merely flexible to stretchable[10–13], intrinsically stretchable OPVs (IS-OPVs) have gained considerable attention regarding their structural configurability, omnidirectional stretchability, and in-plane deformability[14–16]. To date, three approaches have been adopted to realise stretchability: incorporating an additional elastomer component into the active layer, developing new active materials, and enhancing the interfacial adhesion between the transparent electrode and stretchable substrate. Initially, conventional active layer systems such as polymer donor:small-molecule acceptor were modified with a small amount of an additional elastomer to reduce their stiffness and enhance their ductility[17,18]. Subsequently, researchers focused on developing new active materials, in particular, by modifying the polymer chains of conjugated polymer donors to achieve high PCEs as well as superior mechanical properties[15,16,19,20]. Thereafter, in addition to enhancing the intrinsic stretchability of only active layers, achieving strong interfacial adhesion between the component layers was targeted as an effective approach to suppress delamination, delocalise the strain in the top layers, and considerably delay crack initiation and propagation[14,21–23]. Without these approaches, cracks appear and proliferate rapidly in pseudo-freestanding thin films subjected to considerably small tensile strains[16,18,19,21,22,24–26]. Accordingly, a tightly bonded substrate/transparent electrode platform was established with robust adhesion of a poly(3,4-ethylenedioxythiophene) polystyrene sulfonate (PEDOT:PSS) electrode onto a stretchable thermoplastic polyurethane (PU) substrate through a combination of physical and chemical adsorption. The improved adhesion considerably suppressed crack initiation and allowed the IS-OPVs to maintain 79.7% of their initial PCE under 40% strain[14]. Moreover, the aforementioned strategies can be combined to enhance stretchability. For instance, state-of-the-art research has unveiled a novel thymine-sidechain-terminated 6,7-difluoro-quinoxaline monomer, which was used to synthesise a new conjugated polymer donor. Owing to the robust bonding between the substrate and transparent electrode, the resulting thymine-functionalised-terpolymer-based IS-OPVs exhibited a record-high initial PCE of 13.7%, showed substantially improved stretchability, and maintained 80% of the initial PCE under 43% tensile strain[15].

However, devising an approach applicable to IS-OPVs with different active systems while ensuring mechanical robustness against both repetitive strain cycles and high tensile strains remains a formidable challenge. In particular, the most difficult problem is to realise the stretchability and cyclic durability of every single layer in IS-OPVs, specifically the transparent electrode and fragile active layers. Indeed, the tensile modulus of PEDOT:PSS is as high as 2 GPa[27], and plasticiser-incorporated PEDOT:PSS is prone to exhibiting irreversible (plastic) deformation and local delamination from stretchable substrates[28–30], leading to permanent elongation and impaired cycling durability. Furthermore, in general, the high-efficiency polymer donor:small-molecule acceptor active systems can hardly meet the demands for mechanical durability since they exhibit a typical crack onset strain (COS) below 5%[16,18,22,24–26].

In this study, we developed a delocalisation and redistribution strategy for IS-OPVs with improved endurance to high strains and cyclic stretching durability while maintaining a high initial PCE. Using this approach and a new active system, we achieved high-performance IS-OPVs exhibiting an initial PCE of 14.2% and exceptional stretchability, with 80% of the initial PCE ($PCE_{80\%}$) maintained at 52% tensile strain and 95% of the initial PCE retained after 100 stretching cycles at 10% strain. The stretchability was primarily achieved by delocalising

and redistributing the strain in the active layer to the underlying layers, benefiting from the improved stretchability of the PEDOT:PSS electrode realised by incorporating the zwitterion 4-(3-ethyl-1-imidazolio)−1-butanesulfonate (ION E) additive[31]. ION E substantially enhanced the stretchability of PEDOT:PSS by tuning its crystalline structure and strengthening the interfacial adhesion between the PEDOT:PSS layer and PU substrate through reinforced hydrogen bonding. Additionally, we employed a simple-terpolymerisation-based strategy to synthesise a polymer donor, namely Ter-D18, which was then mixed with the small-molecule acceptor Y6, yielding a new active system that helped achieve a high PCE as well as superior mechanical properties. Our redistribution strategy with the new active layer system effectively delayed crack initiation and impeded crack propagation, considerably mitigating the performance degradation of the stretchable OPVs at high tensile strains and under repetitive strain cycles. This device design strategy does not rely solely on the mechanical properties of the active layers to impart stretchability to the entire device, and holds vast potential for extending to various other benchmark active systems, thereby opening a new avenue for the development of IS-OPVs.

## Results

### The configuration and constituent materials of the IS-OPVs

The configuration and constituent materials of the IS-OPVs fabricated in this study are shown in Fig. 1a, and the corresponding chemical structures are presented in Fig. 1b. PU was selected as the substrate owing to its high transparency, high stretchability, and smooth surface[22]. A highly stretchable, transparent, and conductive PEDOT:PSS electrode (Clevios PH1000) was developed by mixing 5 wt% ethylene glycol (EG) and 5 mg mL$^{-1}$ ION E[31]. Another PEDOT:PSS (Clevios P VP AI4083) was used as the hole-transporting layer (HTL). The combination of Ter-D18 and Y6 served as a new active system for achieving both high PCE and superior mechanical properties. The electron-transport-layer-free device structure was adopted from our previous investigations[32,33]. Eutectic gallium−indium (EGaIn) liquid metal was directly coated onto the active layer as the top cathode, which prevented the active layer−EGaIn cathode interface from affecting the device performance.

### Characterisation of conductive PEDOT:PSS

Based on the concept that the strain in the active layer can be redistributed to the underlying PEDOT:PSS HTL and electrode layers[22], and given the reported capacity of the PEDOT:PSS HTL to endure strains surpassing 100%[14], it is considered that enhancing the mechanical durability of the fragile PEDOT:PSS (Clevios PH1000) could be pivotal to achieving high-performance IS-OPVs. Therefore, ION E was chosen as it provides high stretchability for PEDOT:PSS by altering its crystalline structure, as explained later. To that end, PEDOT:PSS (Clevios PH1000) was initially mixed with 5 wt% EG to improve its electrical conductivity and then blended with different amounts of ION E (0, 2, 5, and 10 mg mL$^{-1}$; denoted as 0-ION E, 2-ION E, 5-ION E, and 10-ION E, respectively). These samples are collectively referred to as conductive PEDOT:PSS. First, the viscosities of the conductive PEDOT:PSS solutions were measured with a rheometer. All solutions exhibited shear-thinning behaviour where dynamic viscosity decreased with the applied shear rate, and the viscosities decreased slightly with higher ION E concentrations within the shear rate range of 10 to 1000 s$^{-1}$ (Supplementary Fig. 1). All the conductive PEDOT:PSS films were spin-coated onto glass substrates at 2000 rpm, and their electrical and optical properties were characterised (Supplementary Table 1). The film thickness increased gradually with increasing ION E concentration, from 60 nm for the ION E-free specimen to 113 nm for 10-ION E, which can be attributed to the presence of larger colloidal particles in solutions with higher amounts of ION E, as evidenced by dynamic light scattering analysis (Fig. 2a). Additionally, all the films exhibited similar

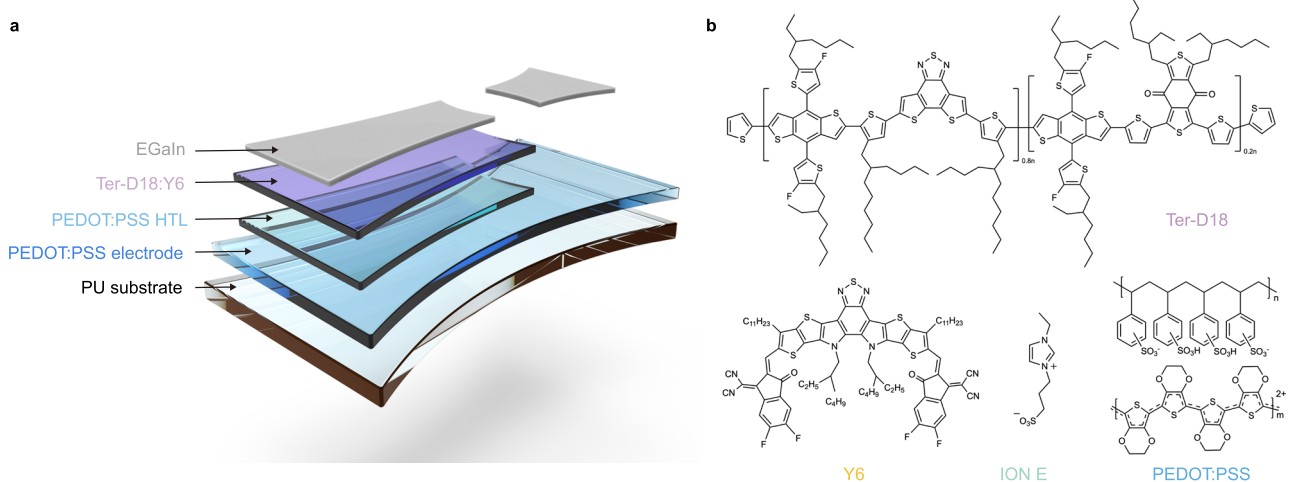

**Fig. 1 | Device structure and materials. a** Schematic of the intrinsically stretchable organic photovoltaics (IS-OPVs) configured as polyurethane (PU)//conductive PEDOT:PSS with 5 mg mL−1 ION E//PEDOT:PSS hole-transporting layer (HTL)//Ter-D18:Y6//eutectic gallium−indium (EGaIn). **b** Chemical structures of the electrode materials and active materials in the IS-OPVs.

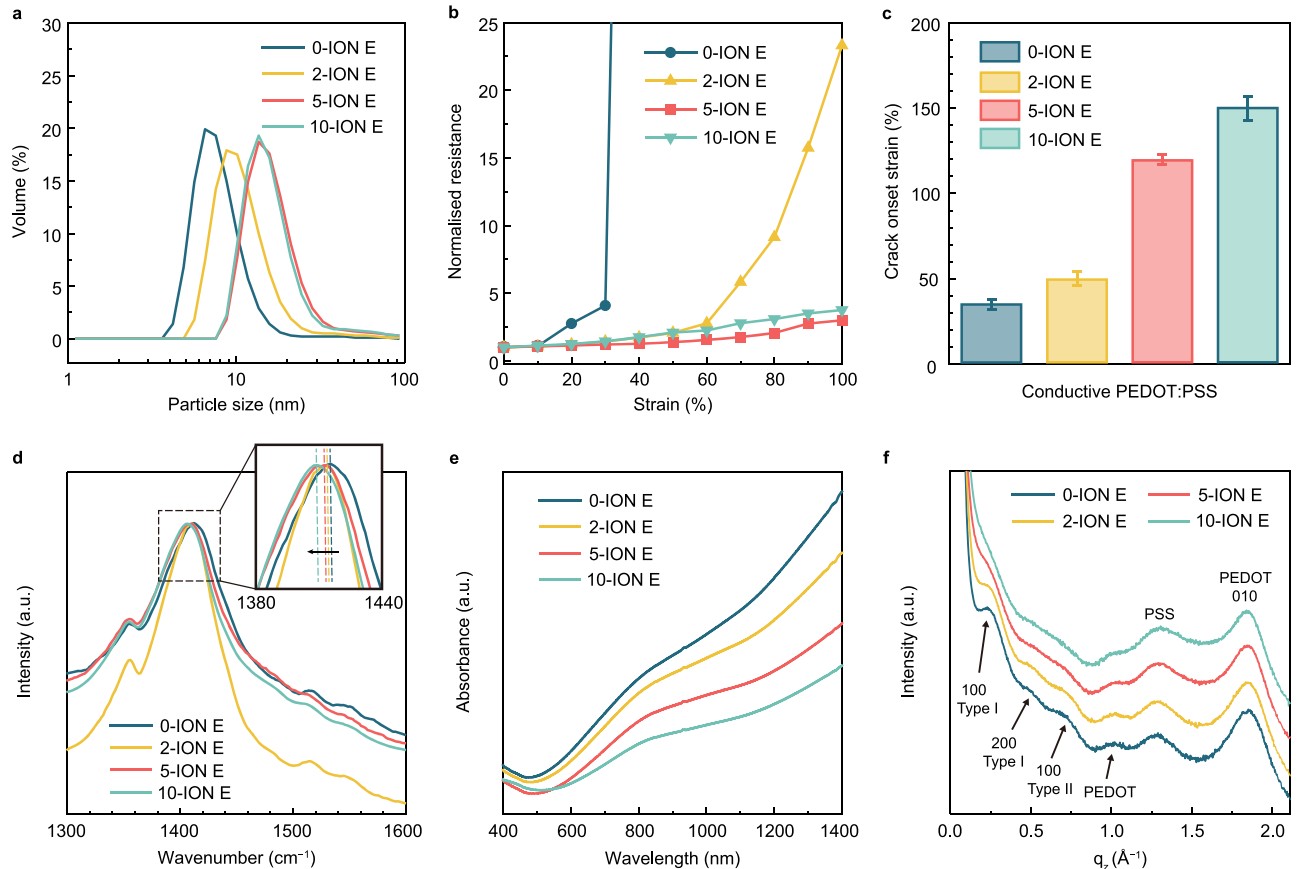

**Fig. 2 | Characterisation of conductive PEDOT:PSS. a** Size distribution of particles in aqueous dispersions of conductive PEDOT:PSS with 0, 2, 5, and 10 mg mL−1 ION E additive. **b** Normalised resistance of PU//conductive PEDOT:PSS films during elongation. **c** Crack onset strain values of PU//conductive PEDOT:PSS films with different amounts of ION E. **d** Raman spectra of PEDOT:PSS electrodes with different amounts of ION E, with an emphasis on the Cα=Cβ peak shift. **e** UV−vis absorption spectra and (**f**) GIXRD patterns of conductive PEDOT:PSS films with different amounts of ION E.

sheet resistances of ~200 Ω/□ and transparency of ~88% at 550 nm (Supplementary Fig. 2). To identify their electrical and mechanical properties under various tensile strains, conductive PEDOT:PSS films were spin coated on 10-µm-thick PU substrates. Notably, the incorporation of ION E considerably mitigated the increase in resistance with increasing tensile strain, with 5-ION E exhibiting a resistance less than twice the initial value at strains of up to 80% (Fig. 2b). In contrast, the ION E-free conductive PEDOT:PSS film exhibited a dramatically increased resistance at 40% strain (122 times higher than the initial value). Such electrical properties correspond to the mechanical

durability of the conductive PEDOT:PSS films. Furthermore, the film stretchability was amplified with increasing ION E concentration (Fig. 2c). The COS values of the films were determined by optical microscopy (OM)-based in situ monitoring (see OM images of the films under various tensile strains in Supplementary Fig. 3). Cracks appeared only in the ION E-free PU//conductive PEDOT:PSS at 35% strain, and they swiftly propagated in the direction perpendicular to the stretching orientation at a slightly higher strain of 40%. The COS values clearly increased with increasing ION E concentration (50%, 120%, and 150% for 2-ION E, 5-ION E, and 10-ION E, respectively). Therefore, crack propagation was significantly inhibited, with substantially shorter and sparser cracks forming under larger tensile strains.

Subsequently, the molecular conformation and crystalline structure of the conductive PEDOT:PSS films were investigated. X-ray photoelectron spectroscopy (XPS) measurements indicated that ION E did not introduce new covalent bonds (Supplementary Figs. 4, 5 and Supplementary Note 1). Raman spectra were acquired to verify the effects of the ION E incorporation on the conformation of PEDOT:PSS[34,35]. The $C_\alpha = C_\beta$ symmetric vibration peaks appeared at ~1412 cm$^{-1}$, and the peaks of the conductive PEDOT:PSS films with higher concentrations of ION E additives were red-shifted; this aligns with the transition of the resonant structure in PEDOT, which shifts from an original benzoid structure to a quinoid configuration[36]. PEDOT with a high quinoid proportion can achieve high planarity, thereby facilitating more efficient carrier delocalisation and the formation of highly ordered stacks of PEDOT chains[37,38]. The oxidation level of PEDOT chains was determined by UV–vis–NIR absorbance spectroscopy of ~90-nm-thick conductive PEDOT:PSS films (Fig. 2e). Three oxidation states of PEDOT chains—neutral, polaron, and bipolaron—correspond to the absorption wavelengths of ~600, ~900, and >1250 nm, respectively, with an elevated absorption intensity within these regions signifying a higher degree of oxidation in the PEDOT chains[35,36,39]. The absorption peak intensity at ~900 and >1250 nm decreased with increasing ION E content, indicating that the inclusion of ION E led to a reduction in the oxidation level of PEDOT. Given that the electrical conductivity of PEDOT is proportional to its oxidation level[40], the conductivity of the conductive PEDOT:PSS films decreased with increasing ION E concentration; this trend is consistent with the regularity of the increased film thickness and consistent sheet resistance (Supplementary Table 1).

Two-dimensional grazing-incidence wide-angle X-ray scattering (GIWAXS) analysis and one-dimensional grazing-incidence X-ray diffraction (GIXRD) (Supplementary Figs. 6, 7 and Fig. 2f, respectively) were performed to explore the crystalline structure of the conductive PEDOT:PSS films incorporated with varying amounts of ION E. Two characteristic peaks appeared at $q_z$ values of ~1.29 Å$^{-1}$ ($d$ = 4.87 Å, calculated using $d = 2\pi/q_z$) and ~1.84 Å$^{-1}$ ($d$ = 3.41 Å), evidently owing to the amorphous stacking of PSS and π–π stacking of PEDOT, respectively[35]; these peaks were almost identical for all the conductive PEDOT:PSS films. Interestingly, the crystal orientation peak of the PEDOT 100 Type I crystal plane appeared at $q_z$ = ~0.23 Å$^{-1}$ ($d$ = 27.32 Å)[35] for the ION E-free conductive PEDOT:PSS film, diminished for the conductive PEDOT:PSS films with 2-ION E and 5-ION E, and almost disappeared for the conductive PEDOT:PSS film with 10-ION E (Fig. 2f). Similar trends were exhibited by the peaks of the PEDOT 200 crystal plane at $q_z$ = ~0.50 Å$^{-1}$ [41] and PEDOT 100 Type II crystal plane at $q_z$ = ~0.69 Å$^{-1}$ [35], as well as the assigned PEDOT peak at $q_z$ = ~0.99 Å$^{-1}$ [41]. The weakened scattering signal intensity implied that the presence of ION E disrupted the highly ordered crystalline configuration of PEDOT:PSS (lamellar stacking) by uncoupling the alternating stacks[42]. This uncoupling effect might stem from the introduction of a new counter-ion exchange between ION E with PEDOT$^+$PSS$^-$, which could shield the strong Coulombic attraction between PSS$^-$ and PEDOT$^+$ as illustrated in the following paragraph[43,44]. Notably combining these results with those of Raman analysis indicated that, on a molecular scale, the ION

E-incorporated PEDOT tended to assume a quinoid backbone structure, making it easier to form π–π stacking compared to that in the absence of ION E. Therefore, the presence of a suitably ordered multiscale morphology guaranteed the high conductivity of the PEDOT:PSS electrode. Furthermore, high stretchability was achieved owing to the disruption of the highly ordered crystalline structures. Because the concentration of PEDOT:PSS in the aqueous solution of Clevios PH1000 is 1.3 wt%[45], the mass ratios of ION E to PEDOT:PSS increased significantly from 0.15 with 2-ION E to 0.77 with 10-ION E. We hypothesise that ION E molecules initially reside between PEDOT and PSS, then occupy the lamellar region of PEDOT domains, and finally disperse into the disordered regions[41]. As illustrated in supplementary Fig. 8, the inserted ION E not only led to the larger particle size as presented in Fig. 2a but also further diluted the PEDOT:PSS concentration in the conductive PEDOT:PSS films with higher ION E concentrations, which corresponds to the constant transparency despite an increase in film thicknesses, as indicated in Supplementary Table 1. However, the surface morphology and phase images obtained via atomic force microscopy (AFM) did not show any apparent differences between the conductive PEDOT:PSS films without and with 5-ION E (Supplementary Fig. 9).

## Interfacial adhesion between conductive PEDOT:PSS and PU

In addition to the ION E-based alteration of the molecular conformation and crystalline structure of conductive PEDOT:PSS, the intermolecular interactions within the conductive PEDOT:PSS, as well as those between the conductive PEDOT:PSS and the PU substrate, were scrutinised (Fig. 3a). Apart from the original reported strong interaction between the −SO$_3$H groups of PSS and −NH$_2$ groups (SO(H···N)H) and O atoms (SO(H···O) = C) of PU[46], the hydrogen bonds among PSSH chains[28], and the Coulombic attraction between PSS$^-$ and PEDOT$^+$[47], the incorporation of ION E introduced new interactions with both the PU substrate and PEDOT:PSS. On the one hand, the counter-ion exchange between PEDOT$^+$PSS$^-$ and ION E[47] was revealed by Fourier transform infrared (FTIR) spectroscopy (Supplementary Fig. 10 and Supplementary Note 2). These interactions lowered the strength of electrostatic interaction between PEDOT and PSS and caused a molecular rearrangement[48,49], which is consistent with the findings from GIXRD analysis, and inhibited the hydrogen bond formation among the PSSH chains[28], thus allowing for increased polymer chain mobility and, consequently, improved mechanical flexibility[41]. On the other hand, the formation of additional hydrogen bonds between ION E and the PU substrate was anticipated to bolster the interfacial adhesion between conductive PEDOT:PSS and the PU substrate[50]. This was corroborated by FTIR spectroscopy (Supplementary Fig. 11). The absorption peak appearing at 3320 cm$^{-1}$ in the spectra corresponded to the hydrogen-bonded N–H groups[51,52], and the enhancement in peak intensity with increasing ION E concentration verified the formation of additional hydrogen bonds between ION E and the PU substrate. Additionally, the contact angle of the conductive PEDOT:PSS droplets on the PU surface decreased with increasing ION E concentration, signifying an improvement in wettability with greater incorporation of ION E (Supplementary Fig. 12). To acquire more insights into the interfacial adhesion mediated by intermolecular interactions, the microscopic adhesive performance of the conductive PEDOT:PSS films was investigated by AFM-based force modulation[53,54]. Two-dimensional adhesion maps of the conductive PEDOT:PSS films without and with 5-ION E were acquired (Figs. 3b, e, respectively), and representative force–displacement curves were extracted from these adhesion maps (Figs. 3c, f). The measured adhesion force for ION E-free PEDOT:PSS (1.15 nN) was significantly lower than that for PEDOT:PSS with 5-ION E (5.66 nN). To determine whether this remarkable microscopic adhesion property was reflected at the macroscopic level, the interfacial adhesion was assessed through pull-off tests[54]. Upon applying a vertical mechanical stress perpendicular to the sample surface,

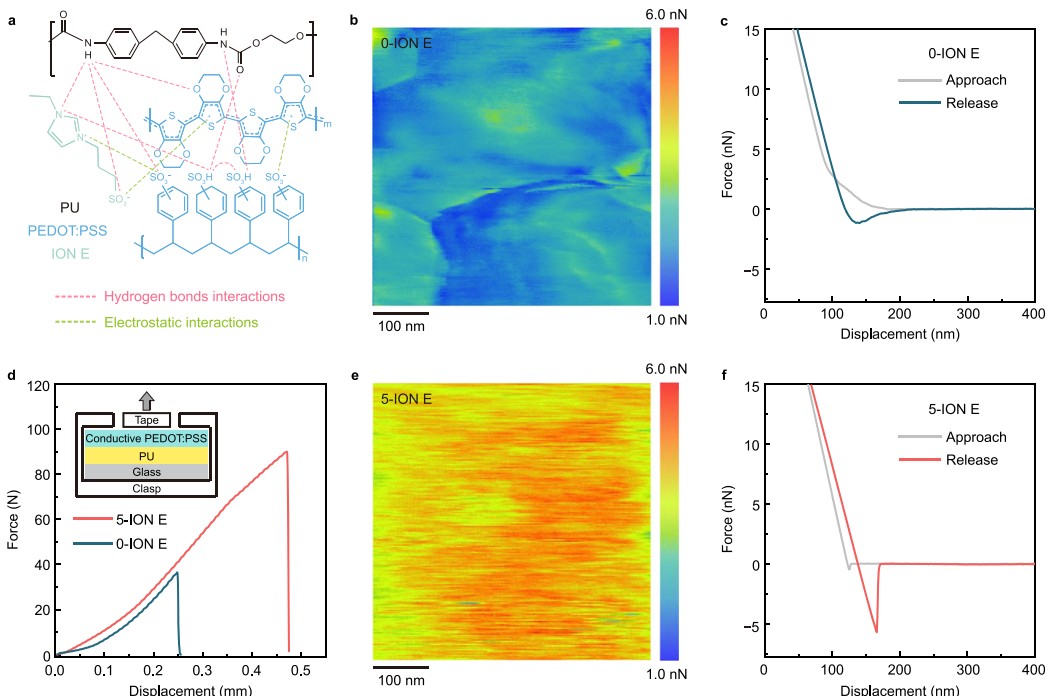

**Fig. 3 | Characterisation of interface between conductive PEDOT:PSS and PU substrate. a** Schematic of interactions enabled by electrostatic forces and hydrogen bonding. **b**, **e** Nanoscale adhesion force maps and (**c**, **f**) AFM-derived force–displacement profiles of conductive PEDOT:PSS (**b**) without ION E and (**e**) with 5 mg mL−1 ION E. **d** Representative force–displacement curves of samples containing the conductive PEDOT:PSS without and with 5 mg mL−1 ION E for characterising adhesion. The inset schematic shows the configuration adopted for the pull-off tests.

delamination occurred at the comparatively weaker interfaces, that is, between conductive PEDOT:PSS and the PU substrate or between the PU layer and the bottom glass substrate. The force–displacement curves obtained through the pull-off tests (Fig. 3d) indicated that the force applied onto the sample with 5-ION E (>80 N) was more than double that onto the ION E-free specimen (see Supplementary Fig. 13 for photos of the samples after the pull-off tests). To confirm the layer delamination, scanning electron microscopy (SEM) was performed in combination with energy-dispersive X-ray spectroscopy (EDX) (see Supplementary Fig. 14 and Supplementary Note 3 for more details). The SEM images revealed a noteworthy enhancement in the interfacial adhesion between the ION E-incorporated conductive PEDOT:PSS film and the PU substrate. It is worth noting that robust interfacial adhesion is crucial for suppressing delamination and dissipating the mechanical stress to the PU layer, which consequently reduces the driving forces for in-plane cracking at the conductive PEDOT:PSS layer, thereby delaying crack initiation and propagation[14,21,55] and imparting enhanced stretchability to the conductive PEDOT:PSS[14].

### Characterisation of the active materials

The stretchability of the active layer is another key aspect to achieve high-performance IS-OPVs. A random terpolymerised polymer donor, namely Ter-D18, was synthesised by incorporating building blocks of two high-efficiency donor materials—PM6 and D18[56]—and the polymerisation conditions were meticulously optimised to obtain a high $M_w$ value (~340 kg mol−1; Supplementary Fig. 15 and Supplementary Table 2). The Ter-D18 donor was mixed with the small-molecule acceptor Y6 to create a new active system for use in this study. First, the performance of OPV devices with commercial indium-tin-oxide anodes was comprehensively characterised (see Supplementary Fig. 16 and Supplementary Note 4 for more details). In addition to the high device performance and good solubility of Ter-D18, its backbone structure was targeted for analysis. The molecular packing behaviours of both Ter-D18 and PM6 were investigated by 2D GIWAXS analysis.

The corresponding line-cut profiles in the pseudo-out-of-plane (OOP) and in-plane (IP) directions are depicted in Supplementary Figs. 17, 18. The results indicated that the terpolymerisation of commonly used building blocks increases the randomness of the backbone structure. The Ter-D18 possesses a weakened crystallinity while enables the formation of a face-on molecular orientation, promoting effective exciton separation and carrier transport (Supplementary Note 5 for more details)[56]. Consequently, all-conjugated terpolymers retain high field-effect mobility while being significantly more stretchable than regular copolymers[57], which was verified by measuring the tensile properties of the pseudo-free-standing active films[58] (See Supplementary Figs. 19 and Supplementary Table. 3). The PM6:Y6 film showed a brittle stress–strain curve without any plastic deformation, exhibiting a COS value of 8.14%. In contrast, the Ter-D18 featured active film showed significantly enhanced stretchability, as indicated by the much higher COS value of 14.81%. The toughness values also improved from 1.43 to 4.30 MJ m−3 for PM6 and Ter-D18 blends, respectively. Therefore, we hypothesised that this simple-terpolymerisation-based design strategy could be leveraged to develop stretchable donor materials for IS-OPVs.

### Strain redistribution

Building upon the promising device performance, the mechanical properties of composite films with the PU//conductive PEDOT:PSS//PEDOT:PSS HTL//active layer configuration were further investigated, and the measurement for PM6:Y6 active system used in our previous studies[32] was also conducted for comparison. The COS values of freestanding composite films (Figs. 4a, d) delaminated from an octadecyltrichlorosilane-modified hydrophobic glass substrate (Figs. 4b, e) were determined by OM. Crack generation was significantly delayed in the composite film with the Ter-D18:Y6 active layer (COS = 60%) compared with that in the film with PM6:Y6, and crack propagation was notably suppressed in the films with 5-ION E and 10-ION E (Figs. 4c, f; Supplementary Figs. 20–22; and Supplementary Note 6). Furthermore, it is worth mentioning that the

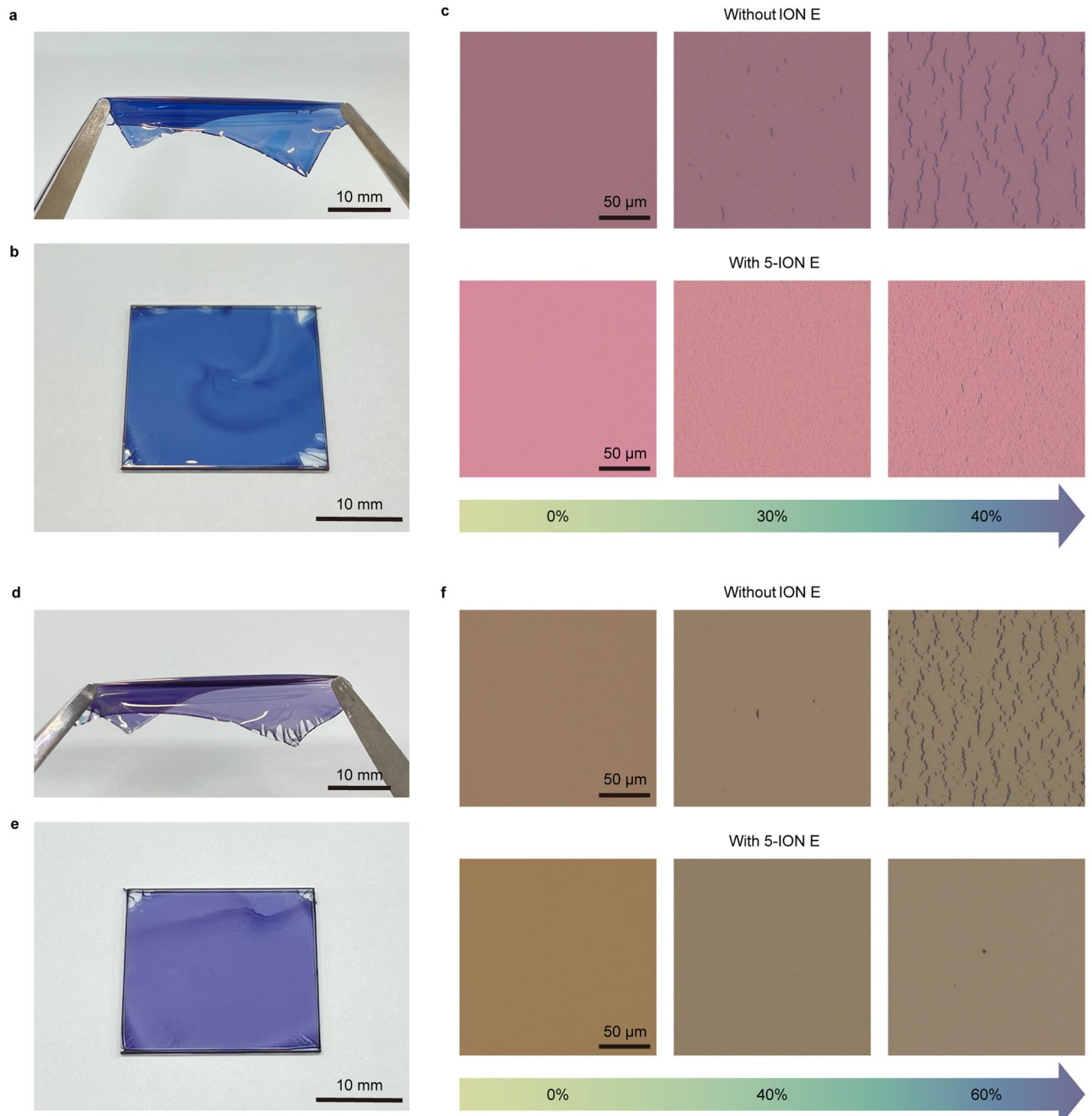

**Fig. 4 | Mechanical properties of PU//conductive PEDOT:PSS//PEDOT:PSS HTL//active layer composite films.** Photographs showing elongation of the (**a**, **b**) PM6:Y6 and (**d**, **e**) Ter-D18:Y6 active layers of freestanding composite films delaminated from an octadecyltrichlorosilane-modified hydrophobic glass substrate. OM images of freestanding composite films with the (**c**) PM6:Y6 and (**f**) Ter-D18:Y6 active systems under tensile strain.

tendency and phenomena of crack propagation in the composite films were similar to that in the PU//conductive PEDOT:PSS films (Supplementary Fig. 3). This resemblance underscores the significant impact of the underlying conductive PEDOT:PSS layer on the uppermost active layer, in that the highly stretchable conductive PEDOT:PSS layer with 5-ION E or 10-ION E effectively delocalised and redistributed the strain in the active layer, thereby delaying crack initiation and propagation, which guaranteed the mechanical integrity of the entire system. It is reported that when a relatively rigid film strongly adheres to a softer underlying layer, the strain will be evenly distributed across the film, so that the film can endure a higher strain than the intrinsic fracture strain of the corresponding free-standing film[59,60]. Accordingly, strong interfacial adherence between the active layers and the underlying PEDOT:PSS layers was necessary to be confirmed through a straightforward 'Scotch tape test' as the prerequisite for efficient strain delocalisation and redistribution (Supplementary Fig. 23)[21]. In addition, we measured the Young's moduli of the conductive PEDOT:PSS films with different ION E concentrations through the buckling-based method[12]. Representative OM images of buckled conductive PEDOT:PSS films with a film thickness of ~100 nm are exhibited in Supplementary Fig. 24, and the Young's moduli extracted from buckling experiments were calculated to be 1.59 GPa, 362 MPa, and 279 MPa for conductive PEDOT:PSS films with 0-ION E, 2-ION E and 5-ION E, respectively (Supplementary Fig. 25). The substantially decreased Young's modulus effectively alleviated the mechanically unstable interface originating from the modulus mismatch between the rigid

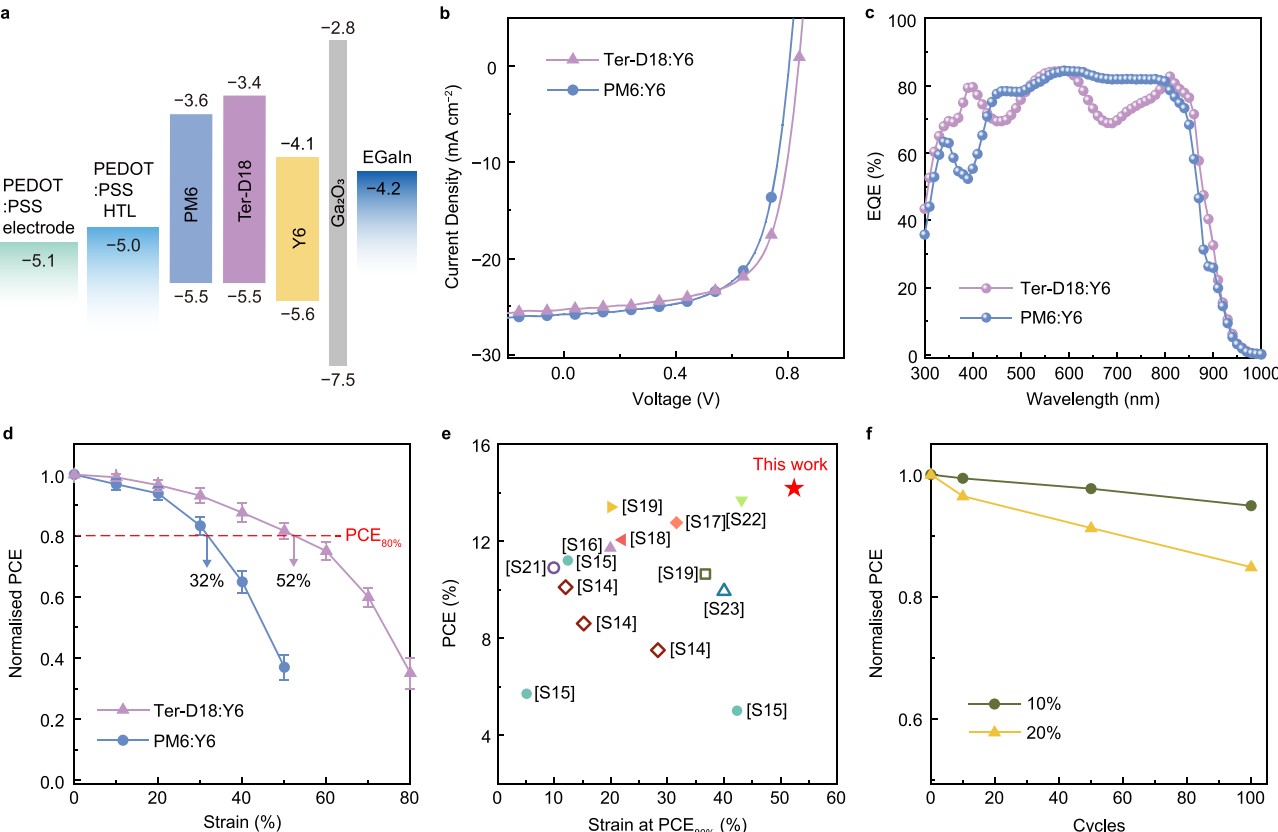

**Fig. 5 | Device performance of IS-OPVs. a** Energy-level diagram of the IS-OPVs. **b** J–V curves of the IS-OPVs prior to stretching. **c** External quantum efficiency (EQE) spectra of the IS-OPVs with Ter-D18:Y6 and PM6:Y6 active layers. **d** Normalised power conversion efficiencies (PCEs) of the IS-OPVs under tensile strain. **e** Stretchability of the IS-OPVs compared with that of previously reported systems, expressed as the initial device performance versus the strain at which 80% of the initial PCE (PCE80%) is retained. The PCE80% values for previously reported devices were estimated by interpolation. **f** Normalised PCE of the IS-OPV with the Ter-D18:Y6 active system under repetitive stretch–release cycles at 10% and 20% tensile strains. The error bars in the plots indicate standard deviations (based on three devices).

active layers and the soft PU substrate[61]. Meanwhile, a reduction in strain generation within the rigid active layer was guaranteed by increasing the ratio of Young's modulus of the top layer to that of the underlying layer[62].

## Performance of the IS-OPV devices

After confirming the exceptional robustness of the composite film under high tensile strains, 5-ION E-incorporated conductive PEDOT:PSS was used as the electrode, and the EGaIn liquid metal cathode was deposited onto the active layer to completely assemble the IS-OPV device (detailed fabrication process was provided in the Experimental Section and Supplementary Fig. 26). The spontaneously formed gallium oxide semiconductor layer with a wide band gap between −7.5 and −2.8 eV[63] and an ultrathin thickness of ~3 nm[64] served as a tunnel, which facilitated efficient charge extraction[63], and thus ensured device performance comparable to those with electron transport layers[65]. By eliminating the need for additional electron interface layers, we effectively prevented any adverse effects on device performance from the active layer–EGaIn cathode interface during the stretching process. Given that liquid metals exhibit ultimate deformability[66], the influence from the EGaIn cathodes on the stretchability of the device is supposed to be negligible. Figure 5a displays the energy-level diagram of the IS-OPVs with the device structure of PU//5-ION E-incorporated conductive PEDOT:PSS//PEDOT:PSS HTL//Ter-D18:Y6 or PM6:Y6 active layer//EGaIn cathode. The work function of the PEDOT:PSS electrode was determined to be 5.13 eV (Supplementary Fig. 27). The IS-OPVs with the Ter-D18:Y6 active system exhibited high initial performance

with a short-circuit current density ($J_{SC}$) of 25.18 mA cm$^{-2}$, open-circuit voltage ($V_{OC}$) of 0.84 V, fill factor (FF) of 0.67, and PCE of 14.18% prior to stretching; these values are superior to those of the PM6:Y6-based IS-OPVs ($J_{SC}$, 25.87 mA cm$^{-2}$; $V_{OC}$, 0.78 V; FF, 0.65; and PCE, 13.14%; Fig. 5b and Supplementary Table 4). The $J_{SC}$ values calculated from the external quantum efficiency spectra (Fig. 5c) are consistent with those obtained from the J–V plots (25.35 and 25.72 mA cm$^{-2}$ for the Ter-D18:Y6- and PM6:Y6-based devices, respectively). The correspondingly reduced initial PCE value of the IS-OPV device compared to the rigid device with ITO electrodes (16.2% in Supplementary Note 4) was mainly due to the lower $J_{SC}$ and FF caused by much larger sheet resistance (~200 Ω/□ vs. ~20 Ω/□ for ITO) and optical loss from the conductive PEDOT:PSS electrodes (Supplementary Fig. 28)[22]. The high sheet resistance exhibits acceptable influence on the device performance when the device area is as small as less than 10 mm$^2$ [67], allowing us to achieve a reasonable high initial PCE.

The stretchable OPV device was subsequently affixed to a stretching stage to analyse its stretchability (Supplementary Fig. 29). To precisely measure the PCE of the IS-OPVs during stretching, a photo mask with an open window of 2 × 2 mm was used to define the same cell area, as illustrated in Fig. 5d and Supplementary Table 5. The PM6:Y6-based IS-OPV maintained 80% of its initial PCE (PCE$_{80\%}$) at a strain of 32% but experienced a rapid deterioration at higher tensile strains. It is worth noting that based on this device design strategy, the stretchability of the IS-OPVs with the unmodified PM6:Y6 active layer surpasses a previously reported value[22]. In sharp contrast, the Ter-D18:Y6-based IS-OPVs exhibited considerably enhanced stretchability,

demonstrating an impressive strain at $PCE_{80\%}$ of 52%. For comparison, even though the devices using 0-ION E conductive PEDOT:PSS electrodes exhibited comparable initial performance to those of the IS-OPVs, they experienced faster degradation (Supplementary Fig. 30 and Supplementary Table 6), and all of them became inoperative under 40% strain due the PEDOT:PSS electrode breakage. The rapid degradation mainly resulted from the limited stretchability of the ION E-free PEDOT:PSS electrodes with swiftly increased sheet resistance, which consequently attributed to a dramatic decrease of FF[68]. In addition, it is worth emphasising that the decrease in PCE during elongation was presumably correlated with crack development. For example, the devices deteriorated gradually before reaching the COS of the composite films, and the primary degradation-inducing factor at this stage was estimated as the thickness reduction of the active layer. As the active layer thinned under tensile strain, the absorbance decreased accordingly[69], and the low absorbance of the thinner active layer led to the loss in $J_{SC}$[14]. The integrity of the entire system was compromised upon reaching the COS, leading to a rapid degradation in PCE at higher tensile strains. This rapid deterioration was a common phenomenon among the IS-OPVs[14,16] and was attributed to synchronised effects from the entire device. First, the generation of cracks introduced "dead areas" in the active layer, which resulted in the vacancy between the electrode and the active layer interface, thus enlarging the contact resistance. Additionally, the permanent cracks produced trap sites, causing electron–hole recombination within the active layers, consequently producing leakage currents and reducing $V_{OC}$[14]. But note that benefiting from the highly stretchable PEDOT:PSS electrode, the cracks propagated slowly and such small cracks likely did not destroy the devices straight away, enabling the high stretchability of the entire device. Furthermore, the IS-OPVs reported herein exhibit a remarkable combination of champion initial performance and stretchability, out-classing all previously reported high-performance IS-OPVs (Fig. 5e and Supplementary Table 7).

Finally, the mechanical durabilities of the IS-OPVs were further investigated under repetitive stretch–release cycles. The resistance evolution of PU//5-ION E-incorporated conductive PEDOT:PSS was first monitored at fixed strains of 10%, 20%, and 30% for 1000 cycles (Supplementary Fig. 31). The specimens subjected to 1000 cycles of 10% and 20% strains exhibited constant resistance, whereas those undergoing 30% strain showed a two-times-higher resistance. Subsequently, the surface morphologies of the PU//5-ION E-incorporated conductive PEDOT:PSS film and the composite films with the Ter-D18:Y6 active layer subjected to 100 cycles of 10% and 20% strains were examined (Supplementary Figs. 32–35 and Supplementary Note 7). According to the results, the Ter-D18:Y6-based IS-OPV maintained 95% and 85% of its initial PCE after 100 cycles of 10% and 20% strains, respectively (Fig. 5f), underlining its comparably remarkable cyclic durability among the previously reported IS-OPVs (Supplementary Table 8).

## Discussion

In this study, we demonstrated the high-performance IS-OPVs exhibiting a remarkable initial PCE of 14.2% with the record high stretchability of $PCE_{80\%} = 52\%$. Moreover, the device showed good mechanical durability, with an impressive 95% of its initial PCE retained after 100 cycles at 10% strain. This excellent device performance was realised by delocalising and redistributing strain to the highly stretchable PEDOT:PSS layer infused with the ION E additive. Moreover, a random terpolymerised polymer donor, namely Ter-D18, was synthesised using a simple terpolymerisation strategy and then mixed with Y6, yielding a new active system for achieving both high PCE and superior mechanical properties. Consequently, this redistribution strategy with a new active layer system effectively suppressed crack initiation and propagation, which considerably mitigated the performance degradation of the stretchable OPVs under high tensile strains and repetitive strain

cycles. Importantly, this device design strategy is not singularly contingent on exploiting the mechanical properties of the active layer to confer stretchability to the entire device, and holds vast potential for probing various other benchmark active systems, thereby charting a new path toward high-performance IS-OPVs fabrication.

## Methods

### Materials

1-Ethylimidazole, 1,3-propanesultone, 2-bromothiophene, and tris(dibenzylideneacetone)dipalladium(0) $(Pd_2(dba)_3)$ were purchased from Tokyo Chemical Industry. 2-(Tributylstannyl)-thiophene was obtained from Sigma-Aldrich. [4,8-Bis[5-(2-ethylhexyl)-4-fluorothiophen-2-yl]-2-trimethylstannylthieno[2,3-f][1]benzothiol-6-yl]-trimethylstannane (BDT-Sn), 1,3-bis(2-ethylhexyl)-5,7-di(bromothiophen-2-yl)benzo[1,2-c:4,5-c′]-dithiophene-4,8-dione (BDD-Br), and dithiophenobenzothiadiazole-c4c8thiophenedibromo (TBT-Br) were procured from Derthon Optoelectronics Materials Science Technology.

Super-dehydrated solvents (toluene and N,N-dimethylformamide (DMF)) and tri(o-tolyl)phosphine $(P(o-tol)3)$ were purchased from FUJIFILM Wako Chemicals. $Pd_2(dba)_3$ was adducted with chloroform to yield $Pd_2(dba)_3CHCl_3$ by recrystallisation, according to a previous report[70]. All the other aforementioned commercial reactants were used without further purification. PM6:Y6 solutions (16.5 mg/mL) were prepared by blending [(2,6-(4,8-bis(5-(2-ethylhexyl-3-fluoro)thiophen-2-yl)-benzo[1,2-b:4,5-b0]dithiophene))-alt-(5,5-(10,30-di-2-thienyl-50,70-bis(2 ethylhexyl) benzo [10,20-c:40,50-c0]dithiophene-4,8-dione))] (PM6; 1-Material) and 2,20-((2Z,20Z)-((12,13-bis(2-ethylhexyl)−3,9-diundecyl-12,13-dihydro [1,2,5]thiadiazolo [3,4-e] thieno[2,"30:4,50]thieno[20,30:4,5]pyrrolo[3,2 g]thieno[20,30:4,5] thieno[3,2-b]indole-2,10-diyl)bis(methanylylidene))bis(5,6-difluoro-3-oxo-2,3-dihydro-1H indene-2,1-diylidene)) dimalononitrile (Y6; 1-Material) in a weight ratio of 5:6 in a solution of chloroform $(CHCl_3;$ FUJIFILM Wako Chemicals) with 0.5% 1-chloronaphthalene (CN) as the additive (Sigma–Aldrich). Two PEDOT:PSS specimens (Clevious P VP Al4083 and Clevious PH1000, respectively) were purchased from Heraeus. Ethylene glycol (EG) was obtained from FUJIFILM Wako Chemicals. Eutectic gallium–indium alloy (EGaIn) was prepared by melting a mixture of gallium (99.99%, Furuuchi Chemical) and indium (99.99%, Furuuchi Chemical) in a weight ratio of 75.5:24.5 at 80 °C for 2 h in a nitrogen glovebox. A glass substrate with a patterned 150-nm-thick indium tin oxide (ITO) electrode was purchased from GEOMA-TEC. A polyurethane (PU) solution (19 wt%) was prepared by diluting a pristine PU solution (Rezamin M-8115P, 30 wt% Dainichiseika) with a mixture of DMF and methyl ethyl ketone (MEK) (wt/wt = 7:3) (FUJIFILM Wako Chemicals).

### Synthesis of 1-Ethyl-3-(3-sulfopropyl)-imidazolium para-toluenesulfonate (ION E)

1-Ethylimidazole (1.1 g, 11.4 mmol) was dissolved in toluene (15 mL) in a pressure-proof vial. 1,3-Propanesultone (1.4 g, 11.4 mmol) was then added to the solution, and the resulting mixture was refluxed and stirred at 110 °C for 19 h. The target zwitterion ION E precipitated as a white solid during this reaction. The precipitate was isolated by filtration and dried under reduced pressure. Quantitative yield, 97%. [1]H NMR (399 MHz, DMSO-$D_6$) δ 9.18 (s, 1H), 7.80 (d, 2H), 4.29 (t, $J = 7.0$ Hz, 2H), 4.19 (q, $J = 7.3$ Hz, 2H), 2.40 (t, $J = 7.2$ Hz, 2H), 2.09 (p, $J = 7.1$ Hz, 2H), 1.42 (t, $J = 0.7$ Hz, 3H) (Supplementary Figs. 36, 37).

### Synthesis of Ter-D18

BDT-Sn (188 mg, 0.2 mmol), TBT-Br (145 mg, 0.16 mmol), and BDD-Br (31 mg, 0.04 mmol) were dissolved in super-dehydrated toluene (10 mL) and DMF (1.1 mL) in a 10–20 mL microwave vial. Moreover, recrystallised $Pd_2(dba)_3CHCl_3$ (5.1 mg, $4.93 \times 10^{-3}$ mmol) and $P(o-tol)_3$ (9 mg, 0.030 mmol) were dissolved in toluene (1.15 mL) in a 3 mL vial,

followed by stirring for 10 min at room temperature. Subsequently, the prepared Pd catalyst solution (1 mL) was added to the reaction vial, which was then closed; all procedures were conducted in a nitrogen-filled glovebox. The reaction vial was loaded into a Biotage® Initiator + Microwave System and subjected to the following procedures: pre-stirring (600 rpm) at room temperature for 30 s and stirring for 2 min (600 rpm) at 100 °C, 2 min (900 rpm) at 120 °C, and 5 h (900 rpm) at 130 °C. Afterwards, for the end-capping reaction, 2-(tri-buthylstannyl)-thiophene was injected into the crude polymer solution, and the resulting mixture was heated at 130 °C for 10 min, followed by the injection of 2-bromothiophene and re-heating at 130 °C for 10 min. After the end-capping reaction, the crude polymer was precipitated from MeOH (250 mL), collected by filtration, and then loaded into a Soxhlet thimble. The thimble was placed inside a Soxhlet extractor, and the crude polymer was successively washed with methanol, acetone, hexane, and dichloromethane, and then extracted with chloroform. Most of the chloroform was evaporated in a rotary evaporator, and the resulting polymer solution was precipitated into methanol. Subsequently, Ter-D18 was collected by filtration and obtained as a black solid (220 mg; yield = 82%; $M_n$ = 89,760; $M_w$ = 339,651; $M_w/M_n$ = 3.78) (Supplementary Fig. 38).

### Preparation of OTS-modified hydrophobic glass substrate
Octadecyltrimethoxysilane (OTMS; 8 µL) was dissolved in tri-chloroethylene (TCE; 5 mL) in a nitrogen glove box. Pre-cleaned glass substrates were treated with oxygen plasma under the following conditions for 5 min: 10 Pa, 5 sccm, 300 W. The prepared solution was spin coated onto the glass substrates at 2000 rpm for 30 s in ambient air. The spin-coated glass substrates were placed on a multilayer shelf, which was subsequently placed in a vacuum bucket after spin coating. Moreover, a 20 mL glass bottle with half-full aqueous ammonia was included, and the bucket was subsequently vacuumed. After storing the resulting samples overnight in the ammonia atmosphere, the glass specimens were removed, washed with methylbenzene by ultra-sonication for 10 min, rinsed with isopropanol, and finally dried for use.

### Device fabrication
A rigid OPV device with the configuration ITO//poly(3,4-ethylene-dioxythiophene):poly(styrenesulfonate) (PEDOT:PSS) (Clevios, P VP AI4083) hole-transporting layer (HTL)//active layer//EGaIn was fabricated as follows: Cr/Au contact pads (3.5-nm-thick Cr and 100-nm-thick Au) were first deposited onto pre-cleaned commercial ITO electrodes via vacuum evaporation. The resulting system was treated with oxygen plasma (10 Pa, 5 sccm, 300 W) for 1 min, following which a 30-nm-thick PEDOT:PSS HTL was spin coated onto the electrode at 4000 rpm for 30 s. The resulting system was annealed at 150 °C on a hot plate for 20 min in ambient air. Next, a Ter-D18:Y6 solution (1:1.2, 14.3 mg/mL, 0.2 wt% CN, in chloroform) was spin coated onto the PEDOT:PSS HTL surface in a nitrogen glovebox, followed by annealing at 110 °C on a hot plate for 5 min. Finally, EGaIn liquid metal was spray coated onto the active area through a shadow mask with an airbrush.

### Fabrication of intrinsically stretchable OPV (IS-OPV) device
An IS-OPV device with the following structure was fabricated: PU//conductive PEDOT:PSS (Clevios PH1000)//PEDOT:PSS HTL//active layer//EGaIn. A 10-µm-thick PU substrate was formed by spin coating 19 wt% PU solution at 2000 rpm for 50 s onto 24 × 24 mm octadecyl-trichlorosilane (OTS)-modified glass to facilitate film release. A conductive PEDOT:PSS solution containing 5 wt% EG and 5 mg mL$^{-1}$ ION E was stirred overnight. Conductive PEDOT:PSS was then spin coated onto the oxygen-plasma-treated PU substrate at 2000 rpm for 50 s, followed by annealing on a hotplate at 110 °C for 10 min in ambient air. Subsequently, Cr/Au contact pads were deposited onto the conductive PEDOT:PSS film via vacuum evaporation, and the conductive

PEDOT:PSS electrode was subsequently patterned via oxygen plasma treatment (10 Pa, 5 sccm, 300 W) for 2 min. Thereafter, the PEDOT:PSS HTL was spin coated onto the electrode at 4000 rpm for 30 s, followed by annealing at 110 °C on a hot plate for 10 min in ambient air. Subsequently, the PM6:Y6 or Ter-D18:Y6 solution was spin coated onto the PEDOT:PSS surface in a nitrogen glovebox, followed by annealing at 110 °C on a hot plate for 5 min. Additional Cr/Au contact pads were deposited onto the active film via vacuum evaporation, and finally, the EGaIn liquid metal was spray coated onto the active area through a shadow mask with an airbrush. The overlap area of the EGaIn electrodes and conductive PEDOT:PSS electrodes are 3.5 × 3.5 mm. After completing the whole fabrication, specially designed Au external wirings were attached to the edge of Au contact pads of the IS-OPVs using an electrically conductive adhesive-transfer tape (3 M, ECATT 9703) to serve as the electrical contacts for measurement during stretching process. The 100 nm thick Au patterns were deposited in a vacuum through a shadow mask onto 12.5 µm thick polyimide films. Double-faced adhesive tapes were then attached to the two edges of the Au external wirings on the IS-OPVs. At last, the device was carefully peeled off from the OTS-modified hydrophobic glass substrate and reversely affixed to a stretching stage. The initial distance between the two edges of the stretching stage was set at 10 mm. The device performance under strains was measured through the connection of the other side of the Au external wirings to the source meter using alligator clips. Additionally, a photo mask with an open window of 2 × 2 mm was covered on the active area to define the same cell area. The detailed fabrication process is illustrated in Supplementary Fig. 26.

### Young's modulus measurement of conductive PEDOT:PSS films
The Young's moduli of conductive PEDOT:PSS films with different concentrations of ION E were measured via the buckling method[27]. First, PDMS (Sylgard 184, Dow Corning) was prepared by mixing the base resin and the curing agent in a weight ratio of 20:1, followed by curing in a 60 °C oven for at least 4 h. Subsequently, the cured PDMS was cut into small rectangular slabs (~2 cm × 5 cm), and the air-side surface of the cured PDMS elastomer was employed for conducting the buckling experiments. The PDMS substrates were first stretched by 2% strain, and then the conductive PEDOT:PSS films were directly spun on them after the oxygen plasma treatment (O₂ 10 sccm, 10 Pa, 100 W, 30 s) to make its surface hydrophilic. The film thickness was controlled by varying the spin speed. The film thicknesses were measured using a surface stylus profilometer (DektakXT, BRUKER). At last, the pre-strained was released and the buckling wavelengths of the PEDOT:PSS films were observed under an optical microscope (VHX-7000, Keyence).

### Pseudo-free-standing tensile test
For the tensile testing specimen, the active layers were spin coated onto the PEDOT:PSS/glass substrate and then etched into a dog-bone sample. To float the specimen on the water surface, water was allowed to penetrate the PEDOT:PSS layer. Subsequently, PEDOT:PSS was dissolved, and the active layer was delaminated from the glass substrate. By performing this process at the water surface, the floating active layer specimen could be obtained. Specimen gripping was achieved by attaching PDMS-coated Al grips on the specimen gripping areas using van der Waals adhesion. The tensile test was performed by a linear stage with a strain rate of 0.002 mm/s. During the tensile test, stress and strain data were obtained through a load cell. All the tensile tests were carried out under ambient conditions (temperature ~25 °C, relative humidity ~30%).

### Characterisation
All the fabricated devices were characterised under simulated solar illumination conditions (AM 1.5 G, 100 mW cm$^{-2}$ (XES-40S3, SAN-EI ELECTRIC) calibrated with a standard silicon reference diode (BS-

520BK, Bunkoukeiki)). *J–V* characteristics were recorded using a Keithley 2400 source meter at a rate of 0.2 V s$^{-1}$ in an ambient atmosphere.

The external quantum efficiency (EQE) was measured using monochromatic light (SM-250F, Bunkoh-Keiki) at wavelengths of 300–1000 nm.

Optical images of the films under different tensile strains were obtained using an optical microscope (VHX-7000, Keyence).

Atomic force microscopy (AFM) images of the conductive PED-OT:PSS films with or without the ION E additive were obtained using a Shimadzu SPM-9700HT scanning probe microscope in phase mode. Adhesion force maps were acquired based on the force modulation model. The cantilever used for characterising the adhesion force had a spring constant of 2 N/m.

Ultraviolet–visible absorption spectra and optical transmittance spectra were acquired using a UV-Visible/NIR Spectrophotometer (V-780, JASCO) in the wavelength range of 300–1200 nm.

Raman spectroscopy was performed using a RAMANtouch ST VIS-NIR-RKL system with a 785 nm laser.

Attenuated total reflectance-Fourier transform infrared (ATR-FTIR) spectroscopy was conducted using an FTIR spectrophotometer (IRSpirit, Shimadzu).

Scanning electron microscopy (SEM) and energy-dispersive X-ray spectroscopy (EDX) images were captured using a Quattro S device (FEI) and a Thermo Fisher Scientific system, respectively. The exposed surfaces of the samples after the pull-off tests were examined by SEM and EDX.

Two-dimensional grazing-incidence wide-angle X-ray scattering (GIWAXS) patterns were recorded using a SmartLab diffractometer (Rigaku, Tokyo, Japan) equipped with a HyPix 6000 detector using Cu Kα radiation ($\lambda = 1.5418$ Å) at an incident angle of 0.3°, and the exposure time was 1 h for each sample. One-dimensional GIXRD patterns were obtained using a SmartLab diffractometer (Rigaku, Tokyo, Japan) with Cu Kα radiation ($\lambda = 1.5418$ Å) at an incidence angle of 0.3°, and the scan step of the detector was 0.1 °/min.

XPS measurements (KRATOS ULTRA 2 [AXIS Supra], Shimadzu) were performed using a monochromatic AlKα source.

Dynamic light scattering analysis was conducted using a Zetasizer device (Malvern Panalytical) with 100-fold-diluted dispersions of conductive PEDOT:PSS with 0, 2, 5, and 10 mg mL$^{-1}$ ION E additive.

The $^1$H nuclei magnetic resonance (NMR) spectrum of ION E was acquired using a JEOL JNM-ECZ400 system operating at 400 MHz for $^1$H NMR at 25 °C, and the residual solvent was used as the internal reference for 1H ($\delta = 7.26$ ppm in CDCl3).

The number- ($M_n$) and weight-average molecular weights ($M_w$) of Ter-D18 were measured using a Tosoh Eco SEC Elite (HLC-8420GPC) device equipped with a refractive index (RI) detector, and polystyrene size-exclusion gel chromatography columns with a chloroform eluent at 40 °C and calibrated with polystyrene standards.

The contact angles of the conductive PEDOT:PSS droplets on the PU surface were measured using a contact angle meter (DMe-211, Kyowa Interface Science).

The viscosities of the conductive PEDOT:PSS solutions were measured with a rotational rheometer (Anton Paar MCR302e) at a constant temperature of 25 °C.

## Pull-off tests

Pull-off tests were performed using a universal tensile tester (EZ-LX, Shimadzu). To that end, conductive PEDOT:PSS films with/without the ION E additive were spin coated onto the unpatterned PU-coated glass substrate. The film stack was clamped by a clasp, and a holder was attached to the conductive PEDOT:PSS top layer using epoxy tape to transfer the mechanical load. The contact area between the holder and device was 5 × 5 mm, and the pulling speed was 1 mm min$^{-1}$.

## Reporting summary

Further information on research design is available in the Nature Portfolio Reporting Summary linked to this article.

## Data availability

Source data are provided with this paper.

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

## Acknowledgements

This study was financially supported by Japan Society for the Promotion of Science under Grants-in-Aid for Scientific Research (S) (No. JP22H04949) and Fund for the Promotion of Joint International Research (International Leading Research) (no. JP22K21343). N.M., T.Y., and K.F acknowledge the support from JST ASPIRE for Rising Scientists (JPMJAP2336). J.W. acknowledges financial support from the China Scholarship Council. N.M. was supported by JST, PRESTO Grant Number JPMJPR20B7, Japan. W.M. and Q.W. thank the support from the National Key Research and Development Program of China (2022YFE0132400), the NSFC (21875182 and 52173023), Key Scientific and Technological Innovation Team Project of Shaanxi Province (2020TD002), 111 project 2.0 (BP0618008), and China National Postdoctoral Program for Innovative Talent Support under Grant BX20220249. The authors would like to extend their gratitude to Prof. Hongzheng Chen and Mr. Xiangjun Zheng of Zhejiang University for fruitful discussions on the COS measurement of active layer films.

## Author contributions

J.W., K.F., and T.S. conceived and designed the study. J.W. characterised the conductive PEDOT:PSS films, and fabricated and characterised the IS-OPVs. Y.O. synthesised and characterised the ION E additive and Ter-D18, and helped characterise the conductive PEDOT:PSS films. N.W. assisted with the dynamic light scattering, Raman, UV–vis, and FTIR analyses, and helped clarify the intermolecular interactions. K.A. and D.H. assisted with the XRD measurements. D.I. and D.H. helped perform SEM imaging and EDX mapping. Q.W. and W.M. assisted with the measurement of the tensile properties of the active films. D.K. provided the octadecyltrichlorosilane-modified hydrophobic glass substrates. S.X. and L.S. instructed the fabrication of the IS-OPVs. B.D. assisted with the AFM measurements. W.W. helped conduct the contact angle measurements. J.W., Y.O., N.M., C.S., T.Y., K.T., T.A., K.F., and T.S. analysed and interpreted the data and drafted the manuscript with inputs from all co-authors.

## Competing interests

The authors declare no competing interests.

## Additional information

[1]Department of Electrical Engineering and Information Systems, The University of Tokyo, Tokyo 113-8656, Japan. [2]RIKEN Center for Emergent Matter Science (CEMS), Saitama 351-0198, Japan. [3]Institute for Chemical and Bioengineering, ETH Zurich, Zurich 8093, Switzerland. [4]Department of Chemistry and Biotechnology, The University of Tokyo, Tokyo 113-8656, Japan. [5]College of Engineering and Applied Sciences, State Key Laboratory of Analytical Chemistry for Life Science, Nanjing University, Nanjing 210046, China. [6]Research Center for Advanced Science and Technology, The University of Tokyo, Tokyo 153-8505, Japan. [7]Institute of Industrial Science, The University of Tokyo, Tokyo 153-8505, Japan. [8]Institute of Engineering Innovation, The University of Tokyo, Tokyo 113-8656, Japan. [9]State Key Laboratory for Mechanical Behaviour of Materials, Xi'an Jiaotong University, Xi'an 710049, China. [10]Thin-Film Device Laboratory, RIKEN, Saitama 351-0198, Japan. ✉e-mail: kenjiro.fukuda@riken.jp; takao.someya@riken.jp

