## [Peer Review File · Nature Communications]

Intrinsically Stretchable Organic Photovoltaics by
Redistributing Strain to PEDOT:PSS with Enhanced
Stretchability and Interfacial AdhesionREVIEWER COMMENTS

Reviewer #1 (Remarks to the Author):

Comments: The authors reported intrinsically stretchable OPVs based on stretchable transparent electrodes. By incorporating zwitterion (ION E) doped PEDOT:PSS with the new Ter-D18:Y6 active layer, the authors demonstrated stretchable OPVs with an initial efficiency of 14.2%, and the devices can maintain 95% of initial efficiency after repeated stretch and release at a strain of 10%. Whereas the performance of the devices is impressive, there are some evident flaws in this manuscript. Hence, I would like to recommend this manuscript for publication if my following concerns are addressed:

1.The authors took up a substantial amount of space to characterizing ION E doped PEDOT:PSS, which has been reported by the previous publication (Nature 600, 246–252 (2021)). In contrast, limited information was provided regarding their newly synthesized active material, Ter-D18. It remains unclear why Ter-D18 is superior to the conventional PM6:Y6 in terms of stretchability. Additionally, important details such as the thickness of the active layer in the device and the tensile modulus of the active layer were not specified. To address these concerns, I suggest that the authors conduct further characterizations of their active layer, particularly focusing on its mechanical properties.

2.The authors claimed that the high stretchability of the OPVs resulted from a redistribution of strain to PEDOT:PSS electrodes, but they didn't mention how they realized such redistribution. Is it achieved by simply adjusting the doping ratio of ION E in PEDOT:PSS? If so, is it due to the variations in modulus of PEDOT:PSS electrodes? It is important for the authors to provide a comprehensive discussion regarding their redistribution strategy.

Reviewer #2 (Remarks to the Author):

The authors propose a delocalization and redistribution strategy for intrinsically stretchable-organic photovoltaics (IS-OPVs) with superior electrical and mechanical properties. The stretchability of PEDOT:PSS was enhanced after incorporating the ION E additive. Additionally, the terpolymerized polymer donor, which exhibits better miscibility with the acceptor, results in outstanding electrical and mechanical properties. By employing ION E-doped PEDOT:PSS and Ter-D18, the authors achieved a power conversion efficiency (PCE) of 14.2% and maintained 80% of the initial PCE under a 52% strain.

This work provides valuable insights into stretchable solar cells with high electrical and mechanical properties with adequate additives. However, zwitterion 4-(3-ethyl-1-imidazolium)-1-butanethanesulfonate (ION E) (Nature, 2021, 600, 246) and terpolymer strategies (Adv. Funct. Mater., 2022, 32, 2203193) used in this work are not novel concepts in this context. Therefore, it is recommended that the authors provide sufficient originality and novelty in fabricating IS-OPVs compared to previous research.

A few comments:

1. After adopting the ION E, Raman measurements indicated a transformation of PEDOT chain from benzoid to quinoid, a structural change is expected to enhance crystallinity and conductivity. However, contrary to this expectation, UV-vis and GIXRD results revealed a decrease in crystallinity and a reduction in conductivity. What might be the reason for the lack of increased crystallinity despite the configuration change?

2. In this work, two strategies were employed to enhance the stretchability of IS-OPV, in terms of PEDOT:PSS and the active layer. A comparison of normalized PCE result for 'PM6:Y6+0 ION E' and 'Ter-D18:Y6+0 ION E' cases could help determine which factor contributes more significantly to improved stretchability in both initial performance and mechanical durability. The authors should explain it.

3. In general, the addition of ionic additives to PEDOT:PSS can lead to the issue of gelation. Does the addition of ION E result in this kind of gelation? Or are there any changes in viscosity after the introduction of ION E?

4. Despite the nearly two-fold difference in thickness between 0-ION E and 10-ION E, there is not a significant difference in transparency. Why?

5. Which factors contribute to achieving a high initial efficiency when introducing ION E to PEDOT:PSS, despite it having a relatively high sheet resistance of 200 ohms/square and low transparency of 88%?

6. In Figure 3a, the authors insist that hydrogen bond interactions between PU-PEDOT, PU-PSS, and PU-ION E. And electrostatic interactions between ION E-PSS, ION E-PEDOT. Are there any data to support this claim?

7. The authors demonstrated a high power conversion efficiency (PCE) even in the absence of an electron transport layer (ETL). Many studies on IS-OPVs employ ETL for efficient charge transport. Please elaborate it.

8. During the spraying of EGaIn, its spontaneous oxidation is inevitable, leading to the formation of an insulating Ga oxide layer. Although this spontaneously formed Ga oxide layer is typically less than 5 nm thick, its impact is considered negligible due to the necessity for charge carriers to tunnel through. Considering its energy level, could the Ga oxide layer potentially serve as an electron transport layer (ETL)?

9. In supplementary Figure 17, the authors presented photographs of IS-OPVs under tensile strain. Additional details for this measurement are necessary, such as how the electrodes were connected in that particular structure.

10. On page 16, there is a typo (PEODT → PEDOT).

11. In Figure 4, the Ter-D18:Y6 sample exhibits no visible cracks under 60% strain. However, Figure 5 shows a sudden decline in power conversion efficiency (PCE) starting from 60% elongation. Sub-micro cracks may be influencing the performance degradation. Further examination at higher microscope ratios is crucial to establish a correlation between the COS of the composite film and PCE degradation.

12. As the COS can vary with the Poisson ratio of the substrate and the sample design, it is imperative to include both the sample size and the initial distance before stretching in the manuscript to accurately represent the experimental conditions and results.

Reviewer #3 (Remarks to the Author):

In the manuscript titled “Intrinsically Stretchable Organic Photovoltaics by Redistributing Strain to PEDOT:PSS Electrodes with Enhanced Stretchability and Interfacial Adhesion” developed intrinsically stretchable organic photovoltaics (IS-OPVs) with an initial power conversion efficiency of 14.2%. These devices exhibit exceptional stretchability, maintaining 80% of the initial efficiency at 52% tensile strain and retaining 95% efficiency after 100 strain cycles at 10%. The stretchability is achieved by redistributing strain in the active layer to a highly stretchable electrode, delaying crack initiation and propagation and minimizing performance degradation under strain.

This work looks very interesting as the authors have achieved high-performance IS-OPVs maintaining mechanical robustness under strain. This is an interesting issue in the recent years for researchers in this area. Unfortunately, the idea of redistributing strain in the active layer to a highly stretchable electrode is well known already (① [22] ACS Energy Lett. 2021, 6, 7, 2512–2518, ② [14] Advanced Energy Materials 2023, 13, 2300544, etc.), and stretchable electrode materials (PEDOT:PSS with additives) used in this manuscript is not special (① [31] Nature 2021, 246-253, ② Advanced Functional Materials 2023, 33, 2212219).

Even if the proposed work could be of great interest for the readers because of high PCE and exceptional stretchability, I'd recommend comprehensive explanations in the novelty of this work thorough review of the manuscript.

Reviewer #1:**Comment #1-0**

The authors reported intrinsically stretchable OPVs based on stretchable transparent electrodes. By incorporating zwitterion (ION E) doped PEDOT:PSS with the new Ter-D18:Y6 active layer, the authors demonstrated stretchable OPVs with an initial efficiency of 14.2%, and the devices can maintain 95% of initial efficiency after repeated stretch and release at a strain of 10%. Whereas the performance of the devices is impressive, there are some evident flaws in this manuscript. Hence, I would like to recommend this manuscript for publication if my following concerns are addressed:

Reply #1-0

We thank the reviewer for their positive comments and evaluation. We have revised the manuscript to resolve the comments raised and incorporate the suggestions provided by the reviewer.

Comment #1-1

The authors took up a substantial amount of space to characterizing ION E doped PEDOT:PSS, which has been reported by the previous publication (Nature 600, 246–252 (2021)). In contrast, limited information was provided regarding their newly synthesized active material, Ter-D18. It remains unclear why Ter-D18 is superior to the conventional PM6:Y6 in terms of stretchability. Additionally, important details such as the thickness of the active layer in the device and the tensile modulus of the active layer were not specified. To address these concerns, I suggest that the authors conduct further characterizations of their active layer, particularly focusing on its mechanical properties.

Reply#1-1

We would like to thank the reviewer for this important comment.

The molecular packing behaviours of both Ter-D18 and PM6 were investigated by 2D GIWAXS analysis. The corresponding line-cut profiles in the pseudo-out-of-plane (OOP) and in-plane (IP) directions are depicted in Supplementary Figs. 17 and 18. The results indicated that the terpolymerisation of commonly used building blocks increases the randomness of the backbone structure. The Ter-D18 possesses a weakened crystallinity while enables the formation of a face-on molecular orientation, promoting effective exciton separation and carrier transport (Supplementary Note V for more details).^{R1} Consequently, all-conjugated terpolymers retain high field-effect mobility while being significantly more stretchable than regular copolymers,^{R2} which was verified by measuring the tensile properties of the pseudo-free-standing active films^{R3} (See Supplementary Figs. 19 and Supplementary Table. 3). The PM6:Y6 film showed a brittle stress-strain curve without any plastic deformation, exhibiting a COS value of 8.14%. In contrast, the Ter-D18 featured active film showed significantly enhanced stretchability, as indicated by the much higher COS value of 14.81%. The toughness values also improved from 1.43 to 4.30 MJ m⁻³ for PM6 and Ter-D18 blends, respectively.

Supplementary Fig. 17. 2D GIWAXS patterns of a) PM6 and b) Ter-D18 films.

Supplementary Fig. 18. line-cut profiles in the a) pseudo-out-of-plane (OOP) and b) in-plane (IP) directions.

Supplementary Note V. 2D GIWAXS analysis of Ter-D18 and PM6 films.

The 1D line-cut profiles are extracted at different azimuth angles β , which is defined as the angle between the q vector and q_z (from 90° to 270° in our system). The 1D in plane (IP) line-cut profile is obtained by calculating the integrated profile from the 2D GIWAXS frame with a polar integration of β from 95° to 105° for the observed q range. The 1D out of plane (OOP) line-cut profile is obtained by calculating the integrated profile from the 2D GIWAXS frame with a polar integration of β from 160° to 170° for the observed q range.^{R4,5}

From 2D GIWAXS patterns of neat films, it is evident that both Ter-D18 and PM6 display a dominant face-on molecular orientation. PM6 shows strong 100 and weak 200 diffraction peaks at 0.28 and 0.61 \AA^{-1} in the IP direction,^{R6} whereas the 100 peak shifted to a larger q_{xy} for Ter-D18, which suggests a reduced d -spacing. The correspondingly enhanced full width at half maximum (FWHM) of 100 peak from 0.085 \AA^{-1} for PM6 to 0.100 \AA^{-1} for Ter-D18 further demonstrated a smaller crystal grain size of Ter-D18 based on Scherrer equation.^{R7} Additionally, in contrast to Ter-D18 that demonstrated only one single primary diffraction peak when q_z is larger than 0.5 \AA^{-1} in the pseudo-OOP direction, PM6 exhibited multiple diffraction peaks at 0.35 , 0.92 and 1.57 \AA^{-1} , indicating the stronger crystallinity. Moreover, both the Ter-D18 and PM6 neat films exhibited a typical 010 peak, which is assigned to the π - π stacking, in the OOP direction located at $q_z = 1.57 \text{ \AA}^{-1}$, corresponding to the same d -spacing value of 4.00 \AA . Similarly, the correspondingly wider FWHM of 0.321 \AA^{-1} for Ter-D18 than 0.221 \AA^{-1} for PM6 demonstrated a smaller crystal grain size of Ter-D18 compared to PM6. The above results show that Ter-D18 possesses a weakened crystallinity while enables the formation of a face-on molecular orientation, promoting effective exciton separation and carrier transport.^{R1}

Figure S19. Stress-strain curves of the pseudo-free-standing PM6:Y6 and Ter-D18:Y6 active films.

Supplementary Table 3. Crack-onset strain and toughness of PM6:Y6 and Ter-D18:Y6 active films.

Active system	Crack-onset strain (COS)	Toughness (MJ m^{-3})
PM6:Y6	8.14	1.43
Ter-D18:Y6	14.81	4.30

Pseudo-free-standing tensile test

For the tensile testing specimen, the active layers were spin-coated onto the PEDOT:PSS/glass substrate and then etched into a dog-bone sample. To float the specimen on the water surface, water was allowed to penetrate the PEDOT:PSS layer. Subsequently, PEDOT:PSS was dissolved, and the active layer was delaminated from the glass substrate. By performing this process at the water surface, the floating active layer specimen could be obtained. Specimen gripping was achieved by attaching PDMS-coated Al grips on the specimen gripping areas using van der Waals adhesion. The tensile test was performed by a linear stage with a strain rate of 0.002 mm/s. During the tensile test, stress and strain data were obtained through a load cell. All the tensile tests were carried out under ambient conditions (temperature ~ 25 °C, relative humidity ~ 30 %).

[Ref. R1] Lu, H. et al. Random Terpolymer Enabling High-Efficiency Organic Solar Cells Processed by Nonhalogenated Solvent with a Low Nonradiative Energy Loss. *Advanced Functional Materials* 32, 2203193 (2022).

[Ref. R2] Mun, J. et al. A design strategy for high mobility stretchable polymer semiconductors. *Nature Communications* 12, 3572 (2021).

[Ref. R3] Wu, Q. et al. High-Performance All-Polymer Solar Cells with a Pseudo-Bilayer Configuration Enabled by a Stepwise Optimization Strategy. *Advanced Functional Materials* 31, 2010411 (2021).

[Ref. R4] Steele, J.A. et al. How to GIWAXS: Grazing Incidence Wide Angle X-Ray Scattering Applied to Metal Halide Perovskite Thin Films. *Advanced Energy Materials* 13, 2300760 (2023).

[Ref. R5] Mahmood, A. & Wang, J.-L. A Review of Grazing Incidence Small- and Wide-Angle X-Ray Scattering Techniques for Exploring the Film Morphology of Organic Solar Cells. *Solar RRL* 4, 2000337 (2020).

[Ref. R6] Zhang, M., Guo, X., Ma, W., Ade, H. & Hou, J. A Large-Bandgap Conjugated Polymer for Versatile Photovoltaic Applications with High Performance. *Advanced Materials* 27, 4655-4660 (2015).

[Ref. R7] He, K., Chen, N., Wang, C., Wei, L. & Chen, J. Method for Determining Crystal Grain Size by X-Ray Diffraction. *Crystal Research and Technology* 53, 1700157 (2018).

We have added the above explanation in the main text (Page 15, Lines 309–324), Experimental Section, and Supplementary Figs. 17–19, Supplementary Table 3 and Supplementary Note V in the supporting information.

We have added References R1 as ref. [56 and S13], R2–R3 as ref. [57–58] and R4–R7 as ref. [S9–S12].

Comment #1-2

The authors claimed that the high stretchability of the OPVs resulted from a redistribution of strain to PEDOT:PSS electrodes, but they didn't mention how they realized such redistribution. Is it achieved by simply adjusting the doping ratio of ION E in PEDOT:PSS? If so, is it due to the variations in modulus of PEDOT:PSS electrodes? It is important for the authors to provide a comprehensive discussion regarding their redistribution strategy.

Reply#1-2

We would like to thank the reviewer for this valuable comment.

It is worth noting that robust interfacial adhesion is crucial for suppressing delamination and dissipating the mechanical stress to the PU layer, which consequently reduces the driving forces for in-plane cracking at the conductive PEDOT:PSS layer, thereby delaying crack initiation and propagation^{R8-10} and imparting enhanced stretchability to the conductive PEDOT:PSS.^{R8}

The efficient strain redistribution primarily relied on the robust bonding between the active layers and the underlying PEDOT:PSS layers, which was confirmed through a straightforward 'Scotch tape test', showing negligible delamination of the active layer from the PEDOT:PSS surface (Supplementary Fig. 23).^{R9} When a relatively rigid film strongly adheres to a softer underlying layer, the strain is evenly distributed across the film, so that the film can endure a higher strain than the intrinsic fracture strain of the corresponding free-standing film.^{R11, 12} Additionally, we measured the Young's moduli of the conductive PEDOT:PSS films with different ION E concentrations through the buckling-based method.^{R13} Representative OM images of buckled conductive PEDOT:PSS films with a film thickness of ~100 nm are exhibited in Supplementary Fig. 24, and the Young's moduli extracted from buckling experiments were calculated to be 1.59 GPa, 362 MPa, and 279 MPa for conductive PEDOT:PSS films with 0-ION E, 2-ION E and 5-ION E, respectively (Supplementary Fig. 25). The substantially decreased Young's modulus effectively alleviated the mechanically unstable interface originating from the modulus mismatch between the rigid active layers and the soft PU substrate.^{R14} Meanwhile, a reduction in strain generation within the rigid active layer was guaranteed by increasing the ratio of Young's modulus of the top layer to that of the underlying layer.^{R15}

Supplementary Fig. 23. Optical images of the scotch tape test. The active layer cannot be transferred to scotch tape, demonstrating robust bonding between the active layer and the underlying PEDOT:PSS layers.

Supplementary Fig. 24. Representative OM images of buckled conductive PEDOT:PSS films with a) 0-ION E, b) 2-ION E and c) 5-ION E with a film thickness of ~100 nm on PDMS substrate.

Supplementary Fig. 25. Experimental data of buckling wavelength of conductive PEDOT:PSS films with a) 0-ION E, b) 2-ION E and c) 5-ION as a function of film thickness.

In buckling-based metrology, a thin film of material is deposited on a soft, compliant elastomeric substrate such as polydimethylsiloxane (PDMS). Under compressive strain, the film buckles, creating a wavy, wrinkled surface. By applying the well-known buckling formulas in conjunction with the measured buckling wavelength and other relevant material properties, the mechanical modulus of the film material can be extracted.^{R16}

The mechanical modulus of film materials can be calculated according to the following equation:

$$d = 2\pi h \left[\frac{(1 - \nu_s^2)E_f}{(1 - \nu_f^2)E_s} \right]^{1/3}$$

where d and h represent the buckling wavelength and the thickness of the upper film, respectively. ν_s and ν_f represent the Poisson's ratio of the PDMS substrate (0.5) and the film material (0.35), respectively. E_s and E_f are the Young's modulus of the PDMS substrate and the film material.

Young's Modulus Measurement of Conductive PEDOT:PSS Films

The Young's moduli of conductive PEDOT:PSS films with different concentrations of ION E were measured via the buckling method.^{R16} First, PDMS (Sylgard 184, Dow Corning) was prepared by mixing the base resin and the curing agent in a weight ratio of 20:1, followed by curing in a 60 °C oven for at least 4 hours. Subsequently, the cured PDMS was cut into small rectangular slabs (~2 cm × 5 cm), and the air-side surface of the cured PDMS elastomer was employed for conducting the buckling experiments. The PDMS substrates were first stretched by 2% strain, and then the conductive PEDOT:PSS films were directly spun on them after the oxygen plasma treatment (O₂ 10 sccm, 10 Pa, 100W, 30 s) to make its surface hydrophilic. The film thickness was controlled by varying the spin speed. The film thicknesses were measured using a surface stylus profilometer (DektakXT, BRUKER). At last, the pre-strained was released

and the buckling wavelengths of the PEDOT:PSS films were observed under an optical microscope (VHX-7000, Keyence).

[Ref. R8] Lee, S. et al. Intrinsically Stretchable Organic Solar Cells without Cracks under 40% Strain. *Advanced Energy Materials*, 2300533 (2023).

[Ref. R9] Kang, J. et al. Tough-interface-enabled stretchable electronics using non-stretchable polymer semiconductors and conductors. *Nature Nanotechnology* 17, 1265-1271 (2022).

[Ref. R10] Lee, J. et al. Interdiffused thermoplastic urethane-PEDOT:PSS bilayers with superior adhesion properties for high-performance and intrinsically-stretchable organic solar cells. *Journal of Materials Chemistry A* 11, 12846-12855 (2023).

[Ref. R11] Li, T. & Suo, Z. Deformability of thin metal films on elastomer substrates. *International Journal of Solids and Structures* 43, 2351-2363 (2006).

[Ref. R12] Li, T. & Suo, Z. Ductility of thin metal films on polymer substrates modulated by interfacial adhesion. *International Journal of Solids and Structures* 44, 1696-1705 (2007).

[Ref. R13] Stafford, C.M. et al. A buckling-based metrology for measuring the elastic moduli of polymeric thin films. *Nature Materials* 3, 545-550 (2004).

[Ref. R14] Kim, Y. et al. A modulus-engineered multi-layer polymer film with mechanical robustness for the application to highly deformable substrate platform in stretchable electronics. *Chemical Engineering Journal* 431, 134074 (2022).

[Ref. R15] Rehman, H.u., Ahmed, F., Schmid, C., Schaufler, J. & Durst, K. Study on the deformation mechanics of hard brittle coatings on ductile substrates using in-situ tensile testing and cohesive zone FEM modelling. *Surface and Coatings Technology* 207, 163-169 (2012).

[Ref. R16] Tahk, D., Lee, H.H. & Khang, D.-Y. Elastic Moduli of Organic Electronic Materials by the Buckling Method. *Macromolecules* 42, 7079-7083 (2009).

We have added the above explanation in the main text (Page 14, Line 293–297 and Page 16–17, Line 345–362), Experimental Section, and Supplementary Figs. 23–25 in the supporting information.

We have added Reference R8–R16 as ref. [14, 21, 55, 59, 60, 12, 61, 62 and S2].

Reviewer #2:**Comment #2-0**

The authors propose a delocalization and redistribution strategy for intrinsically stretchable-organic photovoltaics (IS-OPVs) with superior electrical and mechanical properties. The stretchability of PEDOT:PSS was enhanced after incorporating the ION E additive. Additionally, the terpolymerized polymer donor, which exhibits better miscibility with the acceptor, results in outstanding electrical and mechanical properties. By employing ION E-doped PEDOT:PSS and Ter-D18, the authors achieved a power conversion efficiency (PCE) of 14.2% and maintained 80% of the initial PCE under a 52% strain. This work provides valuable insights into stretchable solar cells with high electrical and mechanical properties with adequate additives. However, zwitterion 4-(3-ethyl-1-imidazolium)-1-butanefulfonate (ION E) (Nature, 2021, 600, 246) and terpolymer strategies (Adv. Funct. Mater., 2022, 32, 2203193) used in this work are not novel concepts in this context. Therefore, it is recommended that the authors provide sufficient originality and novelty in fabricating IS-OPVs compared to previous research.

Reply #2-0

We would like to thank the reviewer for their constructive comments. We agree that originality and novelty were unclear in the previous manuscript.

We would like to emphasize that our research achieved the high-performance intrinsically stretchable OPV benefiting from the development of a highly stretchable PEDOT:PSS electrode with a straightforward incorporation of ION E. The inclusion of the ION E additive simultaneously enhanced the stretchability of PEDOT:PSS itself and meanwhile reinforced the interfacial adhesion between conductive PEDOT:PSS film and the PU substrate. Both of the enhancements are pivotal factors ensuring the excellent mechanical durability of the PEDOT:PSS electrode. To gain profound insight into the underlying mechanism, our study marks the first systematic exploration to elucidate how ION E enhanced the stretchability of PEDOT:PSS through the comprehensive characterization of the mechanical properties, molecular conformation and crystalline structure after the incorporation of different concentrations of ION E. Additionally, we initially observed the strengthened interfacial adhesion between the ION-E-incorporated conductive PEDOT:PSS film and the PU substrate, and verified in-between intermolecular interactions, consequently facilitating the dissipation of mechanical stress to the PU layer and thereby strengthening the mechanical durability of the conductive PEDOT:PSS electrode. Moreover, as for the active layer, we focused more on the mechanical properties of the terpolymerised polymer donor Ter-D18, rather than its enhanced solubility in nonhalogenated solvents. As a result, both the reduced Young's modulus of the conductive PEDOT:PSS electrode and the reinforced interfacial adhesion played a pivotal role in effectively delocalising and redistributing the strain in the active layer to the underlying layers. Combined with the outstanding performance of the active system with both high PCE and enhanced mechanical durability, these factors collectively contribute to the remarkable stretching performance of the IS-OPVs developed in this study. To conclude, this device design strategy does not rely solely on the mechanical properties of the active layers to impart stretchability to the entire device and holds vast potential for extending to various other benchmark active systems, thereby opening a new avenue for the development of IS-OPVs.

To convey our novelty more clearly, we have revised the abstract accordingly, and have added experiments to observe the decreased Young's modulus with adding ION-E in PEDOT:PSS, and provided more detailed explanation of the strain redistribution.

The stretchability was primarily realised by delocalising and redistributing the strain in the active layer to a highly stretchable PEDOT:PSS electrode developed with a straightforward incorporation of ION E, which simultaneously enhanced the stretchability of PEDOT:PSS itself and meanwhile reinforced the interfacial adhesion with the polyurethane substrate. Both enhancements are pivotal factors ensuring the excellent mechanical durability of the PEDOT:PSS electrode, which further effectively delayed the crack initiation and propagation in the top active layer, and enabled the limited performance degradation under high tensile strains and repetitive strain cycles.

The efficient strain redistribution primarily relied on the robust bonding between the active layers and the underlying PEDOT:PSS layers, which was confirmed through a straightforward 'Scotch tape test', showing negligible delamination of the active layer from the PEDOT:PSS surface (Supplementary Fig. 23).^{R9} When a relatively rigid film strongly adheres to a softer underlying layer, the strain is evenly distributed across the film, so that the film can endure a higher strain than the intrinsic fracture strain of the corresponding free-standing film.^{R11, 12} Additionally, we further measured the Young's moduli of the conductive PEDOT:PSS films with different ION E concentrations through the buckling-based method.^{R13} The representative OM images of buckled conductive PEDOT:PSS films with a film thickness of ~100 nm were exhibited in Supplementary Fig. 24, and the Young's moduli extracted from buckling experiments were calculated to be 1.59 GPa, 362 MPa, and 279 MPa for conductive PEDOT:PSS films with 0-ION E, 2-ION E and 5-ION E, respectively (Supplementary Fig. 25). The substantially decreased Young's modulus effectively alleviated the mechanically unstable interface originating from the modulus mismatch between the rigid active layers and the soft PU substrate.^{R14} Meanwhile, a reduction in strain generation within the rigid active layer was guaranteed by increasing the ratio of Young's modulus of the top layer to that of the underlying layer.^{R15}

Supplementary Fig. 23. Optical images of the scotch tape test. The active layer cannot be transferred to scotch tape, demonstrating robust bonding between the active layer and the underlying PEDOT:PSS layers.

Supplementary Fig. 24. Representative OM images of buckled conductive PEDOT:PSS films with a) 0-ION E, b) 2-ION E and c) 5-ION E with a film thickness of ~100 nm on PDMS substrate.

Supplementary Fig. 25. Experimental data of buckling wavelength of conductive PEDOT:PSS films with a) 0-ION E, b) 2-ION E and c) 5-ION as a function of film thickness.

In buckling-based metrology, a thin film of material is deposited on a soft, compliant elastomeric substrate such as polydimethylsiloxane (PDMS). Under compressive strain, the film buckles, creating a wavy, wrinkled surface. By applying the well-known buckling formulas in conjunction with the measured buckling wavelength and other relevant material properties, the mechanical modulus of the film material can be extracted.^{R16}

The mechanical modulus of film materials can be calculated according to the following equation:

$$d = 2\pi h \left[\frac{(1 - \nu_s^2)E_f}{(1 - \nu_f^2)E_s} \right]^{1/3}$$

where d and h represent the buckling wavelength and the thickness of the upper film, respectively. ν_s and ν_f represent the Poisson's ratio of the PDMS substrate (0.5) and the film material (0.35), respectively. E_s and E_f are the Young's modulus of the PDMS substrate and the film material.

Young's Modulus Measurement of Conductive PEDOT:PSS Films

The Young's moduli of conductive PEDOT:PSS films with different concentrations of ION E were measured via the buckling method.^{R16} First, PDMS (Sylgard 184, Dow Corning) was prepared by mixing the base resin and the curing agent in a weight ratio of 20:1, followed by curing in a 60 °C oven for over 4 hours. Subsequently, the cured PDMS was cut into small rectangular slabs (~2 cm × 5 cm), and the air-side surface of the cured PDMS elastomer was employed for conducting the buckling experiments. The conductive PEDOT:PSS films were directly spun on pre-strained PDMS substrate after the oxygen plasma treatment. (O₂ 10 sccm, 10 Pa, 100W, 30 s) to make its surface hydrophilic. The film thickness was controlled by varying the spin speed. The film thicknesses were measured using a surface stylus profilometer (DektakXT, BRUKER). At last, the pre-strained was released and the buckling wavelengths of the PEDOT:PSS films were observed under an optical microscope (VHX-7000, Keyence).

[Ref. R9] Kang, J. et al. Tough-interface-enabled stretchable electronics using non-stretchable polymer semiconductors and conductors. *Nature Nanotechnology* 17, 1265-1271 (2022).

[Ref. R11] Li, T. & Suo, Z. Deformability of thin metal films on elastomer substrates. *International Journal of Solids and Structures* 43, 2351-2363 (2006).

[Ref. R12] Li, T. & Suo, Z. Ductility of thin metal films on polymer substrates modulated by interfacial adhesion. *International Journal of Solids and Structures* 44, 1696-1705 (2007).

[Ref. R13] Stafford, C.M. et al. A buckling-based metrology for measuring the elastic moduli of polymeric thin films. *Nature Materials* 3, 545-550 (2004).

[Ref. R14] Kim, Y. et al. A modulus-engineered multi-layer polymer film with mechanical robustness for the application to highly deformable substrate platform in stretchable electronics. *Chemical Engineering Journal* 431, 134074 (2022).

[Ref. R15] Rehman, H.u., Ahmed, F., Schmid, C., Schaufler, J. & Durst, K. Study on the deformation mechanics of hard brittle coatings on ductile substrates using in-situ tensile testing and cohesive zone FEM modelling. *Surface and Coatings Technology* 207, 163-169 (2012).

[Ref. R16] Tahk, D., Lee, H.H. & Khang, D.-Y. Elastic Moduli of Organic Electronic Materials by the Buckling Method. *Macromolecules* 42, 7079-7083 (2009).

We have added the above explanation in the abstract (Page 2–3, Line 40–48), main text (Page 16–17, Line 345–362), Experimental Section, and Supplementary Figs. 23–25 in the supporting information.

We have added Reference R9, 11–16 as ref. [21, 59, 60, 12, 61, 62 and S2].

Comment #2-1

After adopting the ION E, Raman measurements indicated a transformation of PEDOT chain from benzoid to quinoid, a structural change is expected to enhance crystallinity and conductivity. However, contrary to this expectation, UV-vis and GIXRD results revealed a decrease in crystallinity and a reduction in conductivity. What might be the reason for the lack of increased crystallinity despite the configuration change?

Reply #2-1

We thank the reviewer for this important comment.

The weakened scattering signal intensity implied that the presence of ION E disrupted the highly ordered crystalline configuration of PEDOT:PSS (lamellar stacking) by uncoupling the alternating stacks.^{R17} This uncoupling effect might stem from the introduction of a new counter-ion exchange between ION E with PEDOT⁺PSS⁻, which could shield the strong Coulombic attraction between PSS⁻ and PEDOT⁺.^{R18, 19}

[Ref. R17] Tu, S. et al. Modulation of electronic and ionic conduction in mixed polymer conductors via additive engineering: Towards targeted applications under varying humidity. *Chemical Engineering Journal* 477, 147034 (2023).

[Ref. R18] Crispin, X. et al. The Origin of the High Conductivity of Poly(3,4-ethylenedioxythiophene)–Poly(styrenesulfonate) (PEDOT–PSS) Plastic Electrodes. *Chemistry of Materials* 18, 4354-4360 (2006).

[Ref. R19] Itoh, K. et al. Structural Alternation Correlated to the Conductivity Enhancement of PEDOT:PSS Films by Secondary Doping. *The Journal of Physical Chemistry C* 123, 13467-13471 (2019).

We have added the above explanation in the main text (Page 11, Lines 224–229).

We have added References R17–R19 as ref. [42–44].

Comment #2-2

In this work, two strategies were employed to enhance the stretchability of IS-OPV, in terms of PEDOT:PSS and the active layer. A comparison of normalized PCE result for 'PM6:Y6+0 ION E' and 'Ter-D18:Y6+0 ION E' cases could help determine which factor contributes more significantly to improved stretchability in both initial performance and mechanical durability. The authors should explain it.

Reply #2-2

We would like to thank the reviewer for this valuable comment.

For comparison, even though the devices using 0-ION E conductive PEDOT:PSS electrodes exhibited comparable initial performance to those of the IS-OPVs, they experienced faster degradation (Supplementary Fig. 30 and Supplementary Table 6), and all of them became inoperative under 40% strain due the PEDOT:PSS electrode breakage. The rapid degradation mainly resulted from the limited

stretchability of the ION E-free PEDOT:PSS electrodes with swiftly increased sheet resistance, which consequently attributed to a dramatic decrease of FF.^{R20}

Supplementary Fig. 30. Normalised power conversion efficiencies (PCEs) of the IS-OPVs using 0-ION E conductive PEDOT:PSS electrodes under tensile strain.

Supplementary Table 6. Photovoltaic parameters of devices using 0-ION E conductive PEDOT:PSS electrodes as a function of strain measured under 100 mW cm⁻² AM 1.5G illumination.

Active system	Strain (%)	J_{sc} (mA cm ⁻²)	V_{oc} (V)	FF	PCE (%)
PM6:Y6	0	25.36	0.79	0.65	13.07
	10	25.04	0.79	0.64	12.49
	20	22.30	0.77	0.57	9.76
	30	17.37	0.67	0.45	5.16
	40	0.00	0.16	0.26	0.00
Ter-D18:Y6	0	24.95	0.83	0.67	13.92
	10	24.74	0.83	0.65	13.32
	20	23.35	0.81	0.61	11.57
	30	21.52	0.77	0.58	9.60
	40	0	0.25	0.26	0.00

[Ref. R20] Qi, B. & Wang, J. Fill factor in organic solar cells. *Physical Chemistry Chemical Physics* 15, 8972-8982 (2013).

We have added above explanation in the main text (Page 19, Lines 405–412), Supplementary Fig. 30, and Supplementary Table 6 in the supporting information.

We have added Reference R20 as ref. [67].

Comment #2-3

In general, the addition of ionic additives to PEDOT:PSS can lead to the issue of gelation. Does the addition of ION E result in this kind of gelation? Or are there any changes in viscosity after the introduction of ION E?

Reply #2-3

We would like to thank the reviewer for this important comment.

First, the viscosities of the conductive PEDOT:PSS solutions were measured with a rheometer. All solutions exhibited shear-thinning behaviour where dynamic viscosity decreased with the applied shear rate, and the viscosities decreased slightly with higher ION E concentrations within the shear rate range of 10 to 1000 s⁻¹ (Supplementary Fig. 1).

Supplementary Fig. 1. The dependency of dynamic viscosity on shear rate for conductive PEDOT:PSS solutions with different amounts of ION E additive.

The viscosities of the conductive PEDOT:PSS solutions were measured with a rotational rheometer (Anton Paar MCR302e) at a constant temperature of 25 °C.

We have added the above explanation in the main text (Page 8, Lines 157–161), Experimental Section, and Supplementary Fig. 1 in the supporting information.

Comment #2-4

Despite the nearly two-fold difference in thickness between 0-ION E and 10-ION E, there is not a significant difference in transparency. Why?

Reply #2-4

We would like to thank the reviewer for this important comment.

Because the concentration of PEDOT:PSS in the aqueous solution of Clevios PH1000 is 1.3 wt%,^{R21} the mass ratios of ION E to PEDOT:PSS increased significantly from 0.15 with 2-ION E to 0.77 with 10-ION E. We hypothesise that ION E molecules initially reside between PEDOT and PSS, then occupy the lamellar region of PEDOT domains, and finally disperse into the disordered regions.^{R22} As illustrated in supplementary Fig. 8, the inserted ION E not only led to the larger particle size as presented in Fig. 2a but also further diluted the PEDOT:PSS concentration in the conductive PEDOT:PSS films with higher ION E concentrations, which corresponds to the constant transparency despite an increase in film thicknesses, as indicated in Supplementary Table 1.

[Ref. R21] Fan, Z., Du, D., Yao, H. & Ouyang, J. Higher PEDOT Molecular Weight Giving Rise to Higher Thermoelectric Property of PEDOT:PSS: A Comparative Study of Clevios P and Clevios PH1000. ACS Applied Materials & Interfaces 9, 11732-11738 (2017).

[Ref. R22] Wang, Y. et al. A highly stretchable, transparent, and conductive polymer. Science Advances 3, e1602076.

Supplementary Fig. 8. Schematic images of the crystalline structures of conductive PEDOT:PSS with different concentrations of ION E.

We have added the above explanation in the main text (Pages 11–12, Lines 236–245) and Supplementary Fig. 8 in the supporting information.

We have added References R21–R22 as ref. [45, 41].

Comment #2-5

Which factors contribute to achieving a high initial efficiency when introducing ION E to PEDOT:PSS, despite it having a relatively high sheet resistance of 200 ohms/square and low transparency of 88%?

Reply #2-5

We would like to thank the reviewer for this important comment.

The correspondingly reduced initial PCE value of the IS-OPV device compared to the rigid device with ITO electrodes (16.2% in Supplementary Note IV) was mainly due to the lower J_{SC} and FF caused by much larger sheet resistance ($\sim 200 \Omega/\square$ vs. $\sim 20 \Omega/\square$ for ITO) and optical loss from the conductive PEDOT:PSS electrodes (Supplementary Fig. 28).^{R23} The high sheet resistance exhibits acceptable influence on the device performance when the device area is as small as less than 10 mm^2 ,^{R24} allowing us to achieve a reasonable high initial PCE.

Supplementary Fig. 28. Transparency of conductive PEDOT:PSS films with 5-ION E and commercial ITO.

[Ref. R23] Noh, J. et al. Intrinsically Stretchable Organic Solar Cells with Efficiencies of over 11%. *ACS Energy Letters* 6, 2512-2518 (2021).

[Ref. R24] Yeo, J.-S. et al. Variations of cell performance in ITO-free organic solar cells with increasing cell areas. *Semiconductor Science and Technology* 26, 034010 (2011).

We have added above explanation in the main text (Page 18, Lines 387–394), and Supplementary Fig. 28 in the supporting information.

We have added References R23–R24 as ref. [22, 66].

Comment #2-6

In Figure 3a, the authors insist that hydrogen bond interactions between PU-PEDOT, PU-PSS, and PU-ION E. And electrostatic interactions between ION E-PSS, ION E-PEDOT. Are there any data to support this claim?

Reply #2-6

We would like to thank the reviewer for the valuable comment.

Apart from the original reported strong interaction between the $-\text{SO}_3\text{H}$ groups of PSS and $-\text{NH}_2$ groups ($\text{SO}(\text{H}\cdots\text{N})\text{H}$) and O atoms ($\text{SO}(\text{H}\cdots\text{O})=\text{C}$) of PU,^{R25} the hydrogen bonds among PSSH chains,^{R26} and the Coulombic attraction between PSS^- and PEDOT^+ .^{R27} the incorporation of ION E introduced new interactions with both the PU substrate and PEDOT:PSS. On the one hand, the counter-ion exchange between $\text{PEDOT}^+\text{PSS}^-$ and $\text{ION E}^{\text{R27}}$ was revealed by Fourier transform infrared (FTIR) spectroscopy (Supplementary Fig. 10 and Supplementary Note II). These interactions lowered the strength of electrostatic interaction between PEDOT and PSS and caused a molecular rearrangement,^{R28, 29} which is consistent with the findings from GIXRD analysis.

Supplementary Fig. 10. FTIR spectra of conductive PEDOT:PSS films with different ION E concentrations.

Supplementary Note II. Interactions between ION E and PEDOT:PSS

The interactions between ION E and PEDOT:PSS were revealed by FTIR spectroscopy. The FTIR bands in the ranges of 1005–1010 cm^{-1} and 1035–1040 cm^{-1} featured the stretching vibrations of the sulfonic group (SO_3H) of PSSH.^{R30} Both FTIR bands blue-shifted in the presence of ION E, indicating a reduction in hydrogen bonds among the PSSH.^{R26} Moreover, the C—O—C stretching band of PEDOT appeared at 1059 cm^{-1} in the absence of 0-ION E and shifted to 1062 cm^{-1} after the incorporation of 5-ION E, which is also indicative of the interaction between PEDOT:PSS and ION E.^{R27}

[Ref. R25] Kim, Y., Yoo, S. & Kim, J.-H. in *Polymers*, Vol. 14 (2022).

[Ref. R26] He, H. et al. *Biocompatible Conductive Polymers with High Conductivity and High Stretchability*. *ACS Applied Materials & Interfaces* 11, 26185-26193 (2019).

[Ref. R27] He, H. et al. *Salt-induced ductilization and strain-insensitive resistance of an intrinsically conducting polymer*. *Science Advances* 8, eabq8160.

[Ref. R28] Kee, S. et al. *Controlling Molecular Ordering in Aqueous Conducting Polymers Using Ionic Liquids*. *Advanced Materials* 28, 8625-8631 (2016).

[Ref. R29] Kim, N. et al. *Elastic conducting polymer composites in thermoelectric modules*. *Nature Communications* 11, 1424 (2020).

[Ref. R30] Khong, S.H. et al. *General Photo-Patterning of Polyelectrolyte Thin Films via Efficient Ionic Bis (Fluorinated Phenyl Azide) Photo-Crosslinkers and their Post-Deposition Modification*. *Advanced Functional Materials* 17, 2490-2499 (2007).

We have added the above explanation in the main text (Page 12, Lines 252–261), and Supplementary Fig. 10 and Supplementary Note II in the supporting information.

We have added Reference R25 as ref. [46], R26 as ref. [28 and S6], R27 as ref. [47 and S7], R28–29 as ref. [48–49] and R30 as ref. [S5].

Comment #2-7

The authors demonstrated a high-power conversion efficiency (PCE) even in the absence of an electron transport layer (ETL). Many studies on IS-OPVs employ ETL for efficient charge transport. Please elaborate it.

Reply #2-7

We would like to thank the reviewer for the valuable comment.

The spontaneously formed gallium oxide semiconductor layer with a wide band gap between -7.5 and -2.8 eV^{R31} and an ultrathin thickness of ~ 3 nm^{R32} served as a tunnel, which facilitated efficient charge extraction,^{R31} and thus ensured device performance comparable to those with electron transport layers. By eliminating the need for additional electron interface layers, we effectively prevented any adverse effects on device performance from the active layer–EGaIn cathode interface during the stretching process.

[Ref. 31] Kim, J.-H. et al. Liquid Metal-Based Perovskite Solar Cells: In Situ Formed Gallium Oxide Interlayer Improves Stability and Efficiency. *Advanced Functional Materials* 13, 2311597 (2023).

[Ref. 32] Ren, L. et al. Nanodroplets for Stretchable Superconducting Circuits. *Advanced Functional Materials* 26, 8111-8118 (2016).

We have added the above explanation in the main text (Page 17–18, Line 368–374).

We have added References R31–R32 as ref. [63–64].

Comment #2-8

During the spraying of EGaIn, its spontaneous oxidation is inevitable, leading to the formation of an insulating Ga oxide layer. Although this spontaneously formed Ga oxide layer is typically less than 5 nm thick, its impact is considered negligible due to the necessity for charge carriers to tunnel through. Considering its energy level, could the Ga oxide layer potentially serve as an electron transport layer (ETL)?

Reply #2-8

We would like to thank the reviewer for the valuable comment.

The spontaneously formed gallium oxide semiconductor layer with a wide band gap between -7.5 and -2.8 eV^{R31} and an ultrathin thickness of ~ 3 nm^{R32} served as a tunnel, which facilitated efficient charge extraction,^{R31} and thus ensured comparable device performance to those with electron transport layers. By eliminating the need for additional electron interface layers, we effectively prevented any adverse effects on device performance from the active layer–EGaIn cathode interface during the stretching process.

[Ref. 31] Kim, J.-H. et al. Liquid Metal-Based Perovskite Solar Cells: In Situ Formed Gallium Oxide Interlayer Improves Stability and Efficiency. *Advanced Functional Materials*, 2311597 (2023).

[Ref. 32] Ren, L. et al. Nanodroplets for Stretchable Superconducting Circuits. *Advanced Functional Materials* 26, 8111-8118 (2016).

We have added the above explanation in the main text (Pages 17–18, Line 368–374).

We have added References R31–R32 as ref. [63–64].

We have also revised the Energy-level diagram of the IS-OPVs in Fig. 5a and Supplementary Fig. 16a.

Figure 5. Device performance of IS-OPVs. a) Energy-level diagram of the IS-OPVs. b) J - V curves of the IS-OPVs prior to stretching. c) External quantum efficiency (EQE) spectra of the IS-OPVs with Ter-D18:Y6 and PM6:Y6 active layers. d) Normalised power conversion efficiencies (PCEs) of the IS-OPVs under tensile strain. e) Stretchability of the IS-OPVs compared with that of previously reported systems, expressed as the initial device performance versus the strain at which 80% of the initial PCE ($PCE_{80\%}$) is retained. The $PCE_{80\%}$ values for previously reported devices were estimated by interpolation. f) Normalised PCE of the IS-OPV with the Ter-D18:Y6 active system under repetitive stretch–release cycles at 10% and 20% tensile strains.

Supplementary Fig. 16. Performance of OPV device with the Ter-D18:Y6 active layer. a) Energy-level diagram of OPV device assembled as follows: ITO//PEDOT:PSS HTL//Ter-D18:Y6//EGaIn. b) Absorption spectrum of 98-nm-thick Ter-D18:Y6 active layer. c) J - V curves acquired under AM 1.5G illumination at 100 mW cm^{-2} . d) External quantum efficiency (EQE) spectrum. Dependence of e) short-circuit current and f) open-circuit voltage on light intensity.

Comment #2-9

In supplementary Figure 17, the authors presented photographs of IS-OPVs under tensile strain. Additional details for this measurement are necessary, such as how the electrodes were connected in that particular structure.

Reply #2-9

We agree.

After completing the whole fabrication, specially designed Au external wirings were attached to the edge of Au contact pads of the IS-OPVs using an electrically conductive adhesive-transfer tape (3M, ECATT 9703) to serve as the electrical contacts for measurement during stretching process. The 100 nm thick Au patterns were deposited in a vacuum through a shadow mask onto $12.5 \mu\text{m}$ thick polyimide films. Double-faced adhesive tapes were then attached to the two edges of the Au external wirings on the IS-OPVs. At last, the device was carefully peeled off from the OTS-modified hydrophobic glass substrate and reversely affixed to a stretching stage. The initial distance between the two edges of the stretching stage was set at 10 mm. The device performance under strains was measured through the connection of the other side of the Au external wirings to the source meter using alligator clips. Additionally, a photo mask with an open window of $2 \text{ mm} \times 2 \text{ mm}$ was covered on the active area to define the same cell area. The detailed fabrication process is illustrated in Supplementary Fig. 26.

Supplementary Fig. 26. Schematic of the fabrication process of IS-OPVs. (1) Spin coating of PU substrate on an OTS-modified glass. (2) Spin coating of conductive PEDOT:PSS electrodes and PEDOT:PSS hole transport layer. (3) Deposit Au contact pads for bottom electrodes. (4) Pattern PEDOT:PSS via oxygen plasma treatment. (5) Spin coating of Ter-D18:Y6 active layer and wipe the edge of Au contact pads. (6) Deposit Au contact pads for top electrodes. (7) Spray coating of EGaIn with a shadow mask. (8) Prepare Au external wirings. (9) Attach Au external wirings to the edge of Au contact pads of the IS-OPVs using an electrically conductive adhesive-transfer tape. (10) Affix the IS-OPV on a stretching stage with double-faced adhesive tapes.

Supplementary Fig. 29. Photographs of a–c) the back side and d–f) the front side of an IS-OPV device subjected to different tensile strains.

We have added the additional details in the Experimental Section and Supplementary Figs. 26 in the supporting information. We have also revised Supplementary Fig.29.

Comment #2-10

On page 16, there is a typo (PEODT □ PEDOT).

Reply #2-10

Thank you very much for the careful check.

We have revised the typo in the main text (Page 20, Line 425).

Comment #2-11

In Figure 4, the Ter-D18:Y6 sample exhibits no visible cracks under 60% strain. However, Figure 5 shows a sudden decline in power conversion efficiency (PCE) starting from 60% elongation. Sub-micro cracks may be influencing the performance degradation. Further examination at higher microscope ratios is crucial to establish a correlation between the COS of the composite film and PCE degradation.

Reply #2-11

We would like to thank the reviewer for the valuable comment.

The integrity of the entire system was compromised upon reaching the COS, leading to a rapid degradation in PCE at higher tensile strains. This rapid deterioration was a common phenomenon among the IS-OPVs^{R8, 33} and was attributed to synchronised effects from the entire device. First, the generation of cracks introduced “dead areas” in the active layer, which resulted in the vacancy between the electrode and the active layer interface, thus enlarging the contact resistance. Additionally, the permanent cracks produced trap sites, causing electron–hole recombination within the active layers, consequently producing leakage currents and reducing V_{OC} .^{R33}

Additionally, there were no sub-micro cracks around the visible cracks observed at higher microscope magnifications upon reaching the COS (Supplementary Figs. 22).

Supplementary Fig. 22. High-magnification OM images of freestanding PU//conductive PEDOT:PSS with 5-ION E//Ter-D18:Y6 films under a) 60%, b) 70% and c) 80% strains.

[Ref. R8] Lee, J.-W. et al. Intrinsically Stretchable, Highly Efficient Organic Solar Cells Enabled by Polymer Donors Featuring Hydrogen-Bonding Spacers. *Advanced Materials* 34, 2207544 (2022).

[Ref. R33] Lee, S. et al. Intrinsically Stretchable Organic Solar Cells without Cracks under 40% Strain. *Advanced Energy Materials*, 2300533 (2023).

We have added the above explanation in the main text (Page 20, Lines 417–424), Supplementary Note VI, and Supplementary Fig. 22 and in the supporting information.

We have added References R8 and R33 as ref. [14, 16].

Comment #2-12

As the COS can vary with the Poisson ratio of the substrate and the sample design, it is imperative to include both the sample size and the initial distance before stretching in the manuscript to accurately represent the experimental conditions and results.

Reply #2-12

We would like to thank the reviewer for the important comment.

A 10- μ m-thick PU substrate was formed by spin coating 19 wt% PU solution at 2000 rpm for 50 s onto 24 mm \times 24 mm octadecyltrichlorosilane (OTS)-modified glass to facilitate film release.

The initial distance between the two edges of the stretching stage was set at 10 mm.

We have added the above explanation in the Experimental Section of the Supporting Information.

Reviewer #3:**Comment #3-0**

In the manuscript titled “Intrinsically Stretchable Organic Photovoltaics by Redistributing Strain to PEDOT:PSS Electrodes with Enhanced Stretchability and Interfacial Adhesion” developed intrinsically stretchable organic photovoltaics (IS-OPVs) with an initial power conversion efficiency of 14.2%. These devices exhibit exceptional stretchability, maintaining 80% of the initial efficiency at 52% tensile strain and retaining 95% efficiency after 100 strain cycles at 10%. The stretchability is achieved by redistributing strain in the active layer to a highly stretchable electrode, delaying crack initiation and propagation and minimizing performance degradation under strain.

This work looks very interesting as the authors have achieved high-performance IS-OPVs maintaining mechanical robustness under strain. This is an interesting issue in the recent years for researchers in this area. Unfortunately, the idea of redistributing strain in the active layer to a highly stretchable electrode is well known already (① [22] ACS Energy Lett. 2021, 6, 7, 2512–2518, ② [14] Advanced Energy Materials 2023, 13, 2300544, etc.), and stretchable electrode materials (PEDOT:PSS with additives) used in this manuscript is not special (① [31] Nature 2021, 246-253, ② Advanced Functional Materials 2023, 33, 2212219).

Even if the proposed work could be of great interest for the readers because of high PCE and exceptional stretchability, I'd recommend comprehensive explanations in the novelty of this work thorough review of the manuscript.

Reply #3-0

We would like to thank the reviewer for their constructive comments. We agree that originality and novelty were unclear in the previous manuscript.

We would like to emphasize that our research achieved the high-performance intrinsically stretchable OPV benefiting from the development of a highly stretchable PEDOT:PSS electrode with a straightforward incorporation of ION E. The inclusion of the ION E additive simultaneously enhanced the stretchability of PEDOT:PSS itself and meanwhile reinforced the interfacial adhesion between conductive PEDOT:PSS film and the PU substrate. Both of the enhancements are pivotal factors ensuring the excellent mechanical durability of the PEDOT:PSS electrode. To gain profound insight into the underlying mechanism, our study marked the first systematic exploration to elucidate how ION E enhanced the stretchability of PEDOT:PSS through the comprehensive characterization of the mechanical properties, molecular conformation and crystalline structure after the incorporation of different concentrations of ION E. Additionally, we initially observed the strengthened interfacial adhesion between the ION-E-incorporated conductive PEDOT:PSS film and the PU substrate, and verified in-between intermolecular interactions, consequently facilitating the dissipation of mechanical stress to the PU layer and thereby strengthening the mechanical durability of the conductive PEDOT:PSS electrode. Moreover, as for the active layer, we focused more on the mechanical properties of the terpolymerised polymer donor Ter-D18, rather than its enhanced solubility in nonhalogenated solvents. As a result, both the reduced Young's modulus of the conductive PEDOT:PSS electrode and the reinforced interfacial adhesion played a pivotal role in effectively delocalising and redistributing the strain in the active layer to the underlying layers. Combined with the outstanding performance of the active system with both high PCE and enhanced mechanical durability, these factors collectively contribute to the remarkable stretching performance of the IS-OPVs developed in this study. To conclude, this device design strategy does not rely solely on the mechanical properties of the active layers to impart stretchability to the entire device and holds vast potential for extending to various other benchmark active systems, thereby opening a new avenue for the development of IS-OPVs.

To convey our novelty more clearly, we have revised the abstract accordingly, and have added experiments to observe the decreased Young's modulus with adding ION-E in PEDOT:PSS, and provided more detailed explanation of the strain redistribution.

The stretchability was primarily realised by delocalising and redistributing the strain in the active layer to a highly stretchable PEDOT:PSS electrode developed with a straightforward incorporation of ION E, which simultaneously enhanced the stretchability of PEDOT:PSS itself and meanwhile reinforced the interfacial adhesion with the polyurethane substrate. Both enhancements are pivotal factors ensuring the excellent mechanical durability of the PEDOT:PSS electrode, which further effectively delayed the crack initiation and propagation in the top active layer, and enabled the limited performance degradation under high tensile strains and repetitive strain cycles.

The efficient strain redistribution primarily relied on the robust bonding between the active layers and the underlying PEDOT:PSS layers, which was confirmed through a straightforward 'Scotch tape test', showing negligible delamination of the active layer from the PEDOT:PSS surface (Supplementary Fig. 23).^{R9} When a relatively rigid film strongly adheres to a softer underlying layer, the strain is evenly distributed across the film, so that the film can endure a higher strain than the intrinsic fracture strain of the corresponding free-standing film.^{R11, 12} Additionally, we further measured the Young's moduli of the conductive PEDOT:PSS films with different ION E concentrations through the buckling-based method.^{R13} The representative OM images of buckled conductive PEDOT:PSS films with a film thickness of ~100 nm were exhibited in Supplementary Fig. 24, and the Young's moduli extracted from buckling experiments were calculated to be 1.59 GPa, 362 MPa, and 279 MPa for conductive PEDOT:PSS films with 0-ION E, 2-ION E and 5-ION E, respectively (Supplementary Fig. 25). The substantially decreased Young's modulus effectively alleviated the mechanically unstable interface originating from the modulus mismatch between the rigid active layers and the soft PU substrate.^{R14} Meanwhile, a reduction in strain generation within the rigid active layer was guaranteed by increasing the ratio of Young's modulus of the top layer to that of the underlying layer.^{R15}

Supplementary Fig. 23. Optical images of the scotch tape test. The active layer cannot be transferred to scotch tape, demonstrating robust bonding between the active layer and the underlying PEDOT:PSS layers.

Supplementary Fig. 24. Representative OM images of buckled conductive PEDOT:PSS films with a) 0-ION E, b) 2-ION E and c) 5-ION E with film thickness of ~100 nm on PDMS substrate.

Supplementary Fig. 25. Experimental data of buckling wavelength of conductive PEDOT:PSS films with a) 0-ION E, b) 2-ION E and c) 5-ION E as a function of film thickness.

In buckling-based metrology, a thin film of material is deposited on a soft, compliant elastomeric substrate such as polydimethylsiloxane (PDMS). Under compressive strain, the film buckles, creating a wavy, wrinkled surface. By applying the well-known buckling formulas in conjunction with the measured buckling wavelength and other relevant material properties, the mechanical modulus of the film material can be extracted.^{R16}

The mechanical modulus of film materials can be calculated according to the following equation:

$$d = 2\pi h \left[\frac{(1 - \nu_s^2)E_f}{(1 - \nu_f^2)E_s} \right]^{1/3}$$

where d and h represent the buckling wavelength and the thickness of the upper film, respectively. ν_s and ν_f represent the Poisson's ratio of the PDMS substrate (0.5) and the film material (0.35), respectively. E_s and E_f are the Young's modulus of the PDMS substrate and the film material.

Young's Modulus Measurement of Conductive PEDOT:PSS Films

The Young's moduli of conductive PEDOT:PSS films with different concentrations of ION E were measured via the buckling method.^{R16} First, PDMS (Sylgard 184, Dow Corning) was prepared by mixing the base resin and the curing agent in a weight ratio of 20:1, followed by curing in a 60 °C oven for over 4 hours. Subsequently, the cured PDMS was cut into small rectangular slabs (~2 cm × 5 cm), and the air-side surface of the cured PDMS elastomer was employed for conducting the buckling experiments. The conductive PEDOT:PSS films were directly spun on pre-strained PDMS substrate after the oxygen plasma treatment. (O₂ 10 sccm, 10 Pa, 100W, 30 s) to make its surface hydrophilic. The film thickness was controlled by varying the spin speed. The film thicknesses were measured using a surface stylus profilometer (DektakXT, BRUKER). At last, the pre-strained was released and the buckling wavelengths of the PEDOT:PSS films were observed under an optical microscope (VHX-7000, Keyence).

[Ref. R9] Kang, J. et al. Tough-interface-enabled stretchable electronics using non-stretchable polymer semiconductors and conductors. *Nature Nanotechnology* 17, 1265-1271 (2022).

[Ref. R11] Li, T. & Suo, Z. Deformability of thin metal films on elastomer substrates. *International Journal of Solids and Structures* 43, 2351-2363 (2006).

[Ref. R12] Li, T. & Suo, Z. Ductility of thin metal films on polymer substrates modulated by interfacial adhesion. *International Journal of Solids and Structures* 44, 1696-1705 (2007).

[Ref. R13] Stafford, C.M. et al. A buckling-based metrology for measuring the elastic moduli of polymeric thin films. *Nature Materials* 3, 545-550 (2004).

[Ref. R14] Kim, Y. et al. A modulus-engineered multi-layer polymer film with mechanical robustness for the application to highly deformable substrate platform in stretchable electronics. *Chemical Engineering Journal* 431, 134074 (2022).

[Ref. R15] Rehman, H.u., Ahmed, F., Schmid, C., Schaufler, J. & Durst, K. Study on the deformation mechanics of hard brittle coatings on ductile substrates using in-situ tensile testing and cohesive zone FEM modeling. *Surface and Coatings Technology* 207, 163-169 (2012).

[Ref. R16] Tahk, D., Lee, H.H. & Khang, D.-Y. Elastic Moduli of Organic Electronic Materials by the Buckling Method. *Macromolecules* 42, 7079-7083 (2009).

We have added the above explanation in the abstract (Pages 2–3, Lines 40–48), main text (Pages 16–17, Lines 345–362), Experimental Section, and Supplementary Figs. 23–25 in the supporting information. We have added References R9, 11–16 as ref. [21, 59, 60, 12, 61, 62 and S2].

REVIEWER COMMENTS

Reviewer #1 (Remarks to the Author):

The authors have well addressed the comments in the revised manuscript. I recommend the publication of this study.

Reviewer #2 (Remarks to the Author):

The authors propose a delocalization and redistribution strategy for intrinsically stretchable-organic photovoltaics (IS-OPVs) with superior electrical and mechanical properties. The stretchability of PEDOT:PSS was enhanced after incorporating the ION E additive. Additionally, the terpolymerized polymer donor, which exhibits better miscibility with the acceptor, results in outstanding electrical and mechanical properties. IS-OPVs with ION E-doped PEDOT:PSS and Ter-D18, the authors achieved a power conversion efficiency (PCE) of 14.2% and maintained 80% of the initial PCE under a 52% strain.

The authors have revised the manuscript as per the comments. The quality of the manuscript has been improved, but it still lacks enough innovation to be published in the high-impact journal Nature Communications.

A few further comments for authors:

1. The authors insist that the inclusion of the ION E additive simultaneously enhanced the stretchability of PEDOT:PSS itself and reinforced the interfacial adhesion between conductive PEDOT:PSS film and PU substrate. However, in supplementary Fig. 23, the authors demonstrated the interfacial adhesion between PEDOT:PSS and the active layer, which is not relevant to their original claim.
2. In supplementary Fig. 22, the authors mentioned that there were no sub-micro cracks in PU/PEDOT:PSS/Active layer films under 60% strain. If there are no microcracks under, there should be no dead areas or charge recombination, so it seems that efficiency should be well maintained. What could be the reasons for less than 80% of initial efficiency at 60% strain?
3. The ref. 16 on Page 20, line 9 does not seem right, it may should be ref. 14. Please check the reference in the MS.

4. In supplementary Fig. 10 and supplementary Note II, the authors state that the FTIR peak of PEDOT:PSS has been blue-shifted after introducing the ION E. However, it seems that all three peaks have been red-shifted.

Reviewer #3 (Remarks to the Author):

This manuscript reports on the high-performance intrinsically stretchable organic photovoltaics. The stretchability was primarily achieved by delocalizing and redistributing the strain in the active layer to a highly stretchable ION E doped PEDOT:PSS electrode. This electrode was developed by incorporating, which simultaneously enhanced the stretchability of PEDOT:PSS itself and reinforced the interfacial adhesion with the polyurethane substrate. The device allowed the high-power conversion efficiency (PCE) of 14.2% and exceptional stretchability (80% of the initial PCE maintained at 52% tensile strain) and mechanical durability. In these regards, the manuscript is technically and scientifically meaningful and informative to the reader in the stretchable electronics society. The authors have addressed my comments and thus I would like to recommend on publishing the manuscript in the journal.

Reviewer #1:**Comment #1-0**

The authors have well addressed the comments in the revised manuscript. I recommend the publication of this study.

Reply #1-0

We would like to thank the reviewer for their positive evaluation.

Reviewer #2:**Comment #2-0**

The authors propose a delocalization and redistribution strategy for intrinsically stretchable-organic photovoltaics (IS-OPVs) with superior electrical and mechanical properties. The stretchability of PEDOT:PSS was enhanced after incorporating the ION E additive. Additionally, the terpolymerized polymer donor, which exhibits better miscibility with the acceptor, results in outstanding electrical and mechanical properties. IS-OPVs with ION E-doped PEDOT:PSS and Ter-D18, the authors achieved a power conversion efficiency (PCE) of 14.2% and maintained 80% of the initial PCE under a 52% strain.

The authors have revised the manuscript as per the comments. The quality of the manuscript has been improved, but it still lacks enough innovation to be published in the high-impact journal Nature Communications.

Reply #2-0

We would like to thank the reviewer for their constructive comments. We have revised the manuscript to resolve the comments raised and incorporate the suggestions provided by the reviewer.

Comment #2-1

The authors insist that the inclusion of the ION E additive simultaneously enhanced the stretchability of PEDOT:PSS itself and reinforced the interfacial adhesion between conductive PEDOT:PSS film and PU substrate. However, in supplementary Fig. 23, the authors demonstrated the interfacial adhesion between PEDOT:PSS and the active layer, which is not relevant to their original claim.

Reply #2-1

We would like to thank the reviewer for this important comment. Based on the comment, we recognized supplementary Fig. 23 needs more explanation. We have revised the main text accordingly.

It is reported that when a relatively rigid film strongly adheres to a softer underlying layer, the strain will be evenly distributed across the film, so that the film can endure a higher strain than the intrinsic fracture strain of the corresponding free-standing film.^{59,60} Accordingly, strong interfacial adherence between the active layers and the underlying PEDOT:PSS layers was

necessary to be confirmed through a straightforward 'Scotch tape test' as the prerequisite for efficient strain delocalisation and redistribution (Supplementary Fig. 23).²¹

We have added the above explanation in the main text (Pages 16–17, Lines 345–351)

Comment #2-2

In supplementary Fig. 22, the authors mentioned that there were no sub-micro cracks in PU/PEDOT:PSS/Active layer films under 60% strain. If there are no microcracks under, there should be no dead areas or charge recombination, so it seems that efficiency should be well maintained. What could be the reasons for less than 80% of initial efficiency at 60% strain?

Reply #2-2

We would like to thank the reviewer for this valuable comment.

the devices deteriorated gradually before reaching the COS of the composite films, and the primary degradation-inducing factor at this stage was estimated as the thickness reduction of the active layer. As the active layer thinned under tensile strain, the absorbance decreased accordingly,^{R1} and the low absorbance of the thinner active layer led to the loss in J_{SC} .¹⁴

[Ref. R1] Zhu, Q. et al. Intermolecular Interaction Control Enables Co-optimization of Efficiency, Deformability, Mechanical and Thermal Stability of Stretchable Organic Solar Cells. *Small* 17, 2007011 (2021).

We have added the above explanation in the main text (Page 19–20, Line 414–418).

We have added Reference R1 as ref. [68].

Comment #2-3

The ref. 16 on Page 20, line 9 does not seem right, it may should be ref. 14. Please check the reference in the MS.

Reply #2-3

Thank you very much for the careful check.

We have revised the reference in the main text (Page 20, Line 426).

Comment #2-4

In supplementary Fig. 10 and supplementary Note II, the authors state that the FTIR peak of PEDOT:PSS has been blue-shifted after introducing the ION E. However, it seems that all three peaks have been red-shifted.

Reply #2-4

We would like to thank the reviewer for this comment.

The x-axis of supplementary Fig.10 is wave number, and the unit is cm^{-1} . All three peaks shifted to larger wave numbers with higher concentrations of ION E, demonstrating that the FTIR bands blue-shifted.

We have added the above explanation in the Supplementary Note II in the supporting information.

Reviewer #3:

Comment #3-0

This manuscript reports on the high-performance intrinsically stretchable organic photovoltaics. The stretchability was primarily achieved by delocalizing and redistributing the strain in the active layer to a highly stretchable ION E doped PEDOT:PSS electrode. This electrode was developed by incorporating, which simultaneously enhanced the stretchability of PEDOT:PSS itself and reinforced the interfacial adhesion with the polyurethane substrate. The device allowed the high-power conversion efficiency (PCE) of 14.2% and exceptional stretchability (80% of the initial PCE maintained at 52% tensile strain) and mechanical durability. In these regards, the manuscript is technically and scientifically meaningful and informative to the reader in the stretchable electronics society. The authors have addressed my comments and thus I would like to recommend on publishing the manuscript in the journal.

Reply #3-0

We would like to thank the reviewer for their positive evaluation.